# Training Diffusion Language Models for Black-Box Optimization

**Zipeng Sun** [* 1 2]  **Can (Sam) Chen** [* † ‡ 2 3]  **Ye Yuan** [1 2]  **Haolun Wu** [1 2]  **Jiayao Gu** [1 2]  **Chris Pal** [2 4 5]  **Xue Liu** [1 2 6]

## Abstract

We study offline black-box optimization (BBO), aiming to discover improved designs from an offline dataset of designs and labels, a problem common in robotics and DNA with limited labeled samples. While recent work applies autoregressive LLMs to BBO by formatting tasks as natural-language prompts, their left-to-right design generation struggles to capture the strong bidirectional dependencies inherent in design problems. To address this, we propose adapting diffusion LLMs to offline BBO to leverage their bidirectional modeling capabilities. However, a domain gap exists between the natural text pre-training of diffusion LLMs and the heterogeneous signals in BBO (prompts, designs, and labels). To bridge this gap, we construct a unified prompt—response corpus and introduce delimiter tokens to explicitly mark field boundaries for *domain adaptation*. We further propose a two-stage *post-training* framework to align the diffusion LLM generation with high-label designs. The first stage performs supervised fine-tuning on the unified dataset via masked-response prediction, and the second stage adopts reinforcement learning with rewards defined by label improvements. Our method achieves state-of-the-art results on Design-Bench under small-data settings with highly efficient training, requiring only 1.5 H100 GPU hours for discrete tasks. Code for our work is available here.

## 1. Introduction

Across diverse scientific disciplines, including robotic design, DNA synthesis, and materials discovery, researchers aim to create novel designs that maximize performance metrics (Trabucco et al., 2022). Nevertheless, obtaining these property measurements often requires direct experimentation that is both labor-costly and time-intensive (Hamidieh, 2018; Angermueller et al., 2019; Barrera et al., 2016; Sample et al., 2019). This constraint essentially precludes the use of standard iterative online optimization. As an alternative, the community turns to offline black-box optimization (BBO) (Kim et al., 2026), which operates by utilizing an existing, static dataset of design-property pairs to propose superior candidates. The central difficulty in this paradigm is that real-world applications frequently suffer from the lack of labeled data points.

Traditional approaches often rely on training task-specific surrogate models or generative models, but they suffer from epistemic uncertainty arising from the aforementioned limited data coverage (Kim et al., 2026). More recent work reformulates task descriptions and offline datasets as natural-language prompts, enabling LLMs to directly generate candidate designs (Yang et al., 2024; Zhang et al., 2023; Liu et al., 2024; Nie et al., 2024; Veličković et al., 2024; Novikov et al., 2025). However, most of these methods employ autoregressive (AR) LLMs (Brown et al., 2020), which are inherently unidirectional. Many design problems (e.g., DNA sequence generation) exhibit bidirectional dependencies, where each unit can be influenced by both its prefix and suffix, making AR models insufficient to fully capture these interactions during left-to-right generation.

In response, we turn to diffusion LLMs for BBO, leveraging their inherent bidirectional modeling capabilities. A diffusion LLM is trained by progressively masking tokens and learning to reconstruct them, and at inference time, it reverses this process by iteratively denoising a fully masked sequence (Nie et al., 2025; Gong et al., 2025; Arriola et al., 2025). This training paradigm naturally enables diffusion LLMs to capture bidirectional dependencies.

As illustrated in Figure 1(a), the BBO setting naturally involves multiple heterogeneous signals: natural-language prompts (task descriptions and instructions) together with offline designs and their associated labels. This introduces a domain gap, as the diffusion LLM is pretrained solely on natural-language text. To bridge this gap, we construct a unified prompt–response

---

[*]Equal contribution  [†]Project lead.  [‡]Work done independent of the author's position at Amazon AGI. [1]McGill University [2]MILA - Quebec AI Institute [3]Amazon AGI [4]Polytechnique Montreal [5]Canada CIFAR AI Chair [6]Mohamed bin Zayed University of Artificial Intelligence.  Correspondence to: Zipeng Sun <zpointsun@gmail.com>, Can (Sam) Chen <chencan421@gmail.com>.

*Proceedings of the 43[rd] International Conference on Machine Learning*, Seoul, South Korea. PMLR 306, 2026. Copyright 2026 by the author(s).

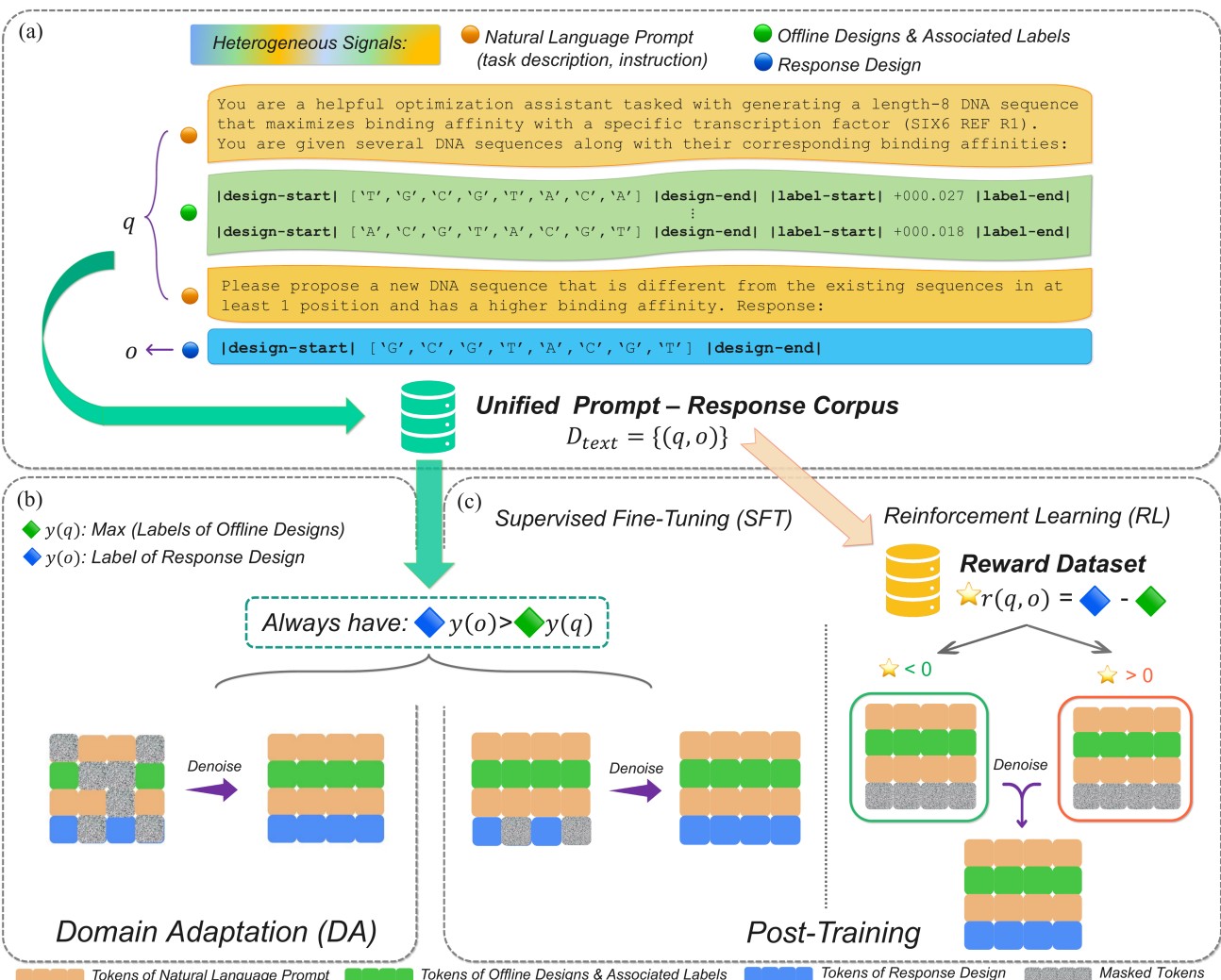

*Figure 1.* **Overview of the DiBO framework.** (a) Unified Prompt–Response Corpus: Heterogeneous BBO signals (natural-language prompts, offline designs and their associated labels) are unified using explicit delimiter tokens. (b) Domain Adaptation (DA): The diffusion LLM is domain-adapted via joint masked-token prediction over prompts and responses. (c) Post-Training: The model is further aligned with high-label designs via supervised fine-tuning (SFT) and reinforcement learning (RL) based on label improvements.

corpus and extend the tokenizer with delimiter tokens—(1) |design-start|/|design-end| to enclose designs and (2) |label-start|/|label-end| to enclose labels, as shown in the middle part of Figure 1 (a). These delimiters explicitly mark the semantic roles of text, design, and label within a unified input sequence. We then perform *domain adaptation* (Figure 1(b)), optimizing the diffusion LLM to jointly predict masked tokens in both the prompt and the response.

We further propose a two-stage *post-training* framework illustrated in Figure 1(c), to align the diffusion LLM's response generation with high-label designs. In the first stage, we perform supervised fine-tuning (SFT) on the unified dataset via masked-response prediction, which instills an inductive bias towards high-label regions of the design space. In the second stage, we construct a reinforcement learning

(RL) dataset where rewards are defined as label improvements from prompt to response, and employ an efficient one-step log-probability approximation to compute the RL loss. This stage further integrates fine-grained reward feedback into the diffusion LLM.

We summarize our main contributions as follows:

- We propose adapting diffusion LLMs to offline BBO to exploit their bidirectional modeling, and construct a unified prompt–response corpus with semantic delimiters to facilitate *domain adaptation*.

- We further propose a two-stage *post-training* framework using SFT and RL to effectively align diffusion generation with high-label designs.

- We demonstrate state-of-the-art performance on Design-Bench in small-data settings.

## 2. Preliminaries and Related Work

### 2.1. Offline Black-Box Optimization

Offline Black-Box Optimization (BBO) aims to identify an optimal design $x^* \in \mathcal{X} \subseteq \mathbb{R}^D$ that maximizes an unknown objective function $f : \mathcal{X} \to \mathbb{R}$ associated with the design $x$:

$$x^* = \arg\max_{x \in \mathcal{X}} f(x). \qquad (1)$$

A practical example involves identifying a DNA sequence to maximize its binding affinity with a specific protein. In many practical scenarios, direct interaction with the objective function is prohibitively expensive or time-consuming. Consequently, we operate under the assumption that we only have access to an offline dataset of previously labeled designs $\mathcal{D} = \{(x_i, y_i)\}_{i=1}^N$, where $y_i = f(x_i)$ (Trabucco et al., 2022; Kim et al., 2026). Our work specifically focuses on small-data settings where the number of labeled samples $N$ is limited (e.g., $N = 500$), reflecting the scarcity of labeled data in real-world applications.

A common baseline is to fit a surrogate model — such as a Deep Neural Network or a Gaussian Process — to approximate $f(\cdot)$ in a supervised manner and then leverage it to guide the design optimization. However, this approach often suffers from the out-of-distribution issue, where the optimizer explores regions of the design space that the surrogate model cannot accurately predict. In this paper, we propose to explore diffusion LLMs due to their robust generalization and bidirectional modeling capabilities.

### 2.2. Diffusion Large Language Models

While autoregressive (AR) models such as GPT (Brown et al., 2020) generate text in a left-to-right way, diffusion LLMs (Nie et al., 2025; Gong et al., 2025) provide a noncausal framework by modeling the data distribution through iterative refinement. This approach offers the distinct advantage of global, bidirectional modeling. Such capabilities are essential in design domains where functional dependencies are often non-sequential; for example, a specific unit within a DNA sequence is constrained not only by its prefix but also by its suffix.

Diffusion LLMs characterize the distribution $p_\theta(x_0)$ via two symmetric processes: forward corruption and reverse reconstruction. Given an initial sequence $x_0$, the forward process introduces noise by stochastically replacing tokens with a [MASK] symbol according to a timestep $t \sim \mathcal{U}[0, 1]$. This yields a partially corrupted sequence $x_t$, which becomes fully masked as $t \to 1$.

The model optimizes a mask predictor $p_\theta(\cdot \mid x_t)$ to recover the original tokens from their corrupted states. The training objective is defined as:

$$\mathcal{L}_{\text{dLLM}} = -\mathbb{E}\left[\frac{1}{t}\sum_{i=1}^L \mathbf{1}[x_t^i = \text{[M]}]\log p_\theta(x_0^i \mid x_t)\right], \quad (2)$$

where $t$ is sampled uniformly from the interval $[0, 1]$, [M] denotes the [MASK] token used for masking, $L$ is the sequence length, and $\theta$ represents the parameters of the diffusion LLM. Once trained, the model generates sequences by starting from total occlusion and iteratively denoising the sequence over a predefined schedule, progressively refining the sequence until convergence.

### 2.3. LLMs for Black-Box Optimization

The expressive capacity of LLMs has recently been leveraged to advance Black-Box Optimization (BBO) (Song et al., 2024). Current research generally bifurcates into two paradigms: (1) **Predictors**, which treat LLMs as surrogate models fine-tuned to estimate the objective value $y$ for a given design $x$ (Raffel et al., 2020; Nguyen et al., 2024; Tan et al., 2025); and (2) **Generators**, which utilize the generative priors of LLMs to directly sample candidate designs $x$ via task-specific prompting (Zhang et al., 2023; Liu et al., 2024; Veličković et al., 2024; Novikov et al., 2025).

Within the generative paradigm, autoregressive (AR) LLMs are typically favored, using task descriptions and collected designs as context for design sampling. For instance, by providing a prompt such as *"Design a DNA sequence with high binding affinity"* alongside offline data, the LLM can be prompted to generate optimized sequences as a response.

While our work follows this generative lineage, we diverge in two fundamental aspects. First, whereas prior efforts rely on AR models constrained by unidirectional causal masking (Zhang et al., 2023; Liu et al., 2024), we employ diffusion LLMs to exploit their inherent bidirectional modeling. This global context is better suited for capturing the intricate structural dependencies common in complex design spaces. Second, we transition from pure prompting to model adaptation. While extant diffusion LLM research (Yuan et al., 2026) is often restricted to frozen-model prompting in few-shot settings, we propose *domain adaptation* and *post-training* on these models. This allows the diffusion LLMs to internalize domain-specific constraints and navigate the optimization landscape effectively, even in data-sparse regimes.

## 3. Methodology

### 3.1. Domain Adaptation

**Heterogeneous Signals** The BBO setting inherently involves heterogeneous inputs: natural-language prompts (task descriptions and instructions) alongside offline de-

signs and their corresponding labels. Since diffusion LLMs are pretrained on natural text, they would otherwise treat designs and labels as ordinary text, leading to ineffective representations and a domain gap. To address this problem, we extend the tokenizer with delimiter tokens: (1) `|design-start|`/`|design-end|` to mark designs, and (2) `|label-start|`/`|label-end|` to enclose label values. These delimiters explicitly define the semantic roles of text, design, and label components within a unified input sequence.

**Unified Corpus** We construct a unified prompt–response corpus that serves as the foundation for both domain adaptation and the subsequent supervised fine-tuning stage. Each input prompt $q$ is formed by concatenating: (1) a task description specifying the semantics, format, and optimization objective of the design and its associated label, and (2) an offline dataset consisting of design–label pairs, followed by an instruction to generate improved designs. The corresponding response $o$ is a design whose label value exceeds all those in the prompt. An example of the final prompt–response format can be found in Figure 1(a). Detailed construction of the dataset $\mathcal{D}_{text} = \{(q, o)\}$ is provided in Appendix A.1 and A.2. Notably, to facilitate effective reasoning by the diffusion LLM, the target response should remain within a reasonable distribution of the prompt examples. Consequently, we select prompt designs based on pairwise kernel similarity to the response, and we elaborate the details in Appendix A.2.

**Joint Prompt–Response Loss** We adopt a joint loss that reconstructs masked tokens in both the prompt $q$ and the response $o$ for domain adaptation:

$$\mathcal{L}_{\text{DA}} = -\mathbb{E}\left[\frac{1}{t}\sum_{i=1}^{L}\mathbf{1}[q_t^i = \text{[M]}, o_t^i = \text{[M]}]\log p_{\boldsymbol{\theta}}(q_0^i, o_0^i \mid q_t, o_t)\right],$$
(3)

where $t$ is sampled uniformly from the interval $[0, 1]$, $L$ is the sequence length, $q_0, o_0$ are clean samples from $\mathcal{D}_{\text{text}}$, $q_t, o_t$ are their masked counterparts, and $\boldsymbol{\theta}$ denotes the parameters of the diffusion LLM. This joint objective enables the model to learn the semantic roles of the delimiter tokens and adapt effectively to the BBO domain context.

### 3.2. Post-Training

We introduce a two-stage *post-training* framework to align diffusion LLMs with high-label design generation.

**Supervised Fine-Tuning** We reuse the unified prompt–response corpus but optimize only over the response sequences, using the following objective:

$$\mathcal{L}_{\text{SFT}} = -\mathbb{E}\left[\frac{1}{t}\sum_{i=1}^{L}\mathbf{1}[o_t^i = \text{[M]}]\log p_{\boldsymbol{\theta}}(o_0^i \mid q_0, o_t)\right].$$
(4)

This supervised fine-tuning stage provides a simple and stable alignment signal that encourages the diffusion LLM to generate improved designs conditioned on the prompt. By reconstructing only masked response tokens, the model learns to map the few-shot prompt context to high-label design outputs. This serves as an effective initialization for the subsequent reinforcement learning stage, which further incorporates fine-grained reward feedback.

**Reward Dataset** We then construct a reinforcement learning dataset $\mathcal{D}_{rl}$, where rewards are defined as label improvements from the prompt to the response:

$$r(q, o) = y(o) - y(q),$$
(5)

where $y(q)$ denotes the highest label among designs appearing in the few-shot prompt. Unlike in SFT, where the response is always constructed to outperform the prompt and thus yields non-negative rewards, the RL phase imposes no such constraint. This construction introduces both positive and negative rewards while preserving local semantic similarity between prompt and response designs, as detailed in Appendix A.2 and A.3.

**Reinforcement Learning** For each pair $(q, o)$, the log probability required by the RL objective is approximated using a one-step unmasking strategy (Zhao et al., 2025) instead of iterative denoising for efficiency and stability:

$$\log p_{\boldsymbol{\theta}}(o \mid q) \approx \sum_{k=1}^{|o|}\log p_{\boldsymbol{\theta}}(o_k \mid q, o_{\text{fullmask}}),$$
(6)

where $o_k$ is the $k$-th token of the response and $o_{\text{fullmask}}$ denotes a fully masked response sequence.

Since the designs in offline BBO are collected from external real-world processes rather than generated by a known language model, the behavior policy (i.e., the old policy) that produced those designs is inaccessible. Following Yan et al. (2025); Fu et al. (2025), we therefore assume this old policy to be uniform. Under this assumption, the denominator of the importance-sampling ratio becomes a constant, and ratio clipping and KL regularization are not applied. The resulting RL objective is:

$$\mathcal{L}_{\text{RL}} = -\mathbb{E}_{q,o}\left[\frac{1}{|o|}\sum_{k=1}^{|o|}p_{\boldsymbol{\theta}}(o_k \mid q, o_{\text{fullmask}})\frac{r(q, o)}{\sigma}\right],$$
(7)

where $(q, o)$ are sampled from $\mathcal{D}_{rl}$, and $\sigma$ denotes the standard deviation of rewards. In this formulation, the term $\frac{r(q,o)}{\sigma}$ serves as the advantage signal. We intentionally omit reward centering (i.e., subtracting the mean) to preserve the inherent prompt-specific information. The RL optimization integrates fine-grained reward feedback into the diffusion LLM, in contrast to the coarse binary supervision in SFT.

# 4. Experiments

We conduct comprehensive experiments to evaluate the effectiveness of our method in offline black-box optimization, with benchmarks detailed in Section 4.1, baselines in Section 4.2, implementation details in Section 4.3, results in Section 4.4, ablations in Section 4.5, and hyperparameter settings described in Section 4.6.

## 4.1. Benchmarks

**Datasets** We consider two discrete sequence design problems and two continuous-parameter tasks drawn from the Design-Bench suite (Trabucco et al., 2022). The *discrete* tasks are: **(1) TF Bind 8 (TF8)** (Barrera et al., 2016), which requires designing an 8-length DNA sequence to maximize binding activity with the SIX6_REF_R1 transcription factor; and **(2) TF Bind 10 (TF10)** (Le et al., 2018), a 10-length DNA sequence design task that optimizes transcription factor binding affinity. Note: The labels of TF Bind 10 in Design-Bench correspond to binding free energy difference (ddG), where lower values indicate stronger binding; we therefore negate the ddG values from the original paper (Le et al., 2018) so that higher scores correspond to stronger binding, consistent with the Design-Bench task description. The *continuous* tasks are: **(3) Ant Morphology (Ant)** (Brockman et al., 2016), which optimizes a 60-dimensional ant body morphology for fast crawling; and **(4) D'Kitty Morphology (D'Kitty)** (Ahn et al., 2020), which optimizes a 56-dimensional D'Kitty robot morphology to navigate toward a fixed target location.

**Evaluation** For all methods, we use the task-specific oracle from the *Design-Bench Benchmark Tasks* (Trabucco et al., 2022) to score generated designs. Following standard practice (Trabucco et al., 2021), each method is allowed to propose 128 candidates per task, and we use the best-achieved value (i.e., the $100^{\text{th}}$ percentile) of the normalized ground-truth score as the main metric. The normalized score $y_n$ is defined as $y_n = \frac{y - y_{\min}}{y_{\max} - y_{\min}}$, where $y$ denotes the raw oracle score of a design, and $y_{\min}, y_{\max}$ are the minimum and maximum scores over the full (unobserved) dataset.

## 4.2. Baselines

We systematically benchmark our approach against a wide range of prior methods.

**Forward Methods** We consider methods that utilize a learned surrogate to guide design optimization. *(1) Grad (mean)* models the black-box function with a Gaussian Process (GP) and performs gradient ascent on the posterior mean to improve existing designs. *(2) Grad (EI)* optimizes the Expected Improvement (EI) acquisition function instead of the mean. For GP baselines, we use the dimension-scaled GP prior configuration of Hvarfner et al. (2024), which is better suited to high-dimensional Bayesian optimization as discussed in Appendix A.4. *(3) COMs* (Trabucco et al., 2021) lower-bounds a neural surrogate's predictions on out-of-distribution inputs and applies gradient ascent on this conservative surrogate. *(4) ICT* (Yuan et al., 2023) maintains three surrogate networks with a rotating pseudo-labeling and co-teaching scheme to mitigate overfitting. *(5) MATCH-OPT* (Hoang et al., 2024) explicitly bounds the mismatch between surrogate gradients and true gradients to improve the quality of surrogate-guided optimization. *(6) UniSO-T* (Tan et al., 2025) models the black-box function as an autoregressive sequence-to-sequence model and uses this learned surrogate to steer design generation.

**Inverse Methods** These methods learn a conditional distribution over high-quality designs using generative models. **VAE:** *(1) CbAS* (Brookes et al., 2019) fits a VAE to the offline data with high-scoring samples emphasized. *(2) ExPT* (Nguyen et al., 2023) fits a Transformer-based VAE on offline data, followed by in-context optimization to sample improved designs. **GAN:** *(3) MIN* (Kumar & Levine, 2020) employs a GAN to model the inverse mapping from scores to designs, conditioning on high scores for direct sampling. **Autoregressive:** *(4) BONET* (Mashkaria et al., 2023) models the trajectory from low- to high-scoring samples with an autoregressive model and unrolls these trajectories at test time to produce new candidates. *(5) OPRO* (Yang et al., 2024) prompts an autoregressive LLM with sequences of past design-label pairs to directly generate new designs. We instantiate OPRO with LLaMA3-8B-Instruct (Grattafiori et al., 2024), whose scale is comparable to our diffusion LLM. **Diffusion:** *(6) GTG* (Yun et al., 2024) trains a conditional diffusion model on synthetic trajectories derived from offline data to guide designs toward high-score regions. *(7) DDOM* (Krishnamoorthy et al., 2023) learns a conditional diffusion model on the offline dataset and employs classifier-free guidance during sampling. *(8) dLLM* (Yuan et al., 2026) directly queries a frozen diffusion LLM with prompts and design-label pairs to generate high-scoring designs via MCTS without fine-tuning the model. Finally, we include *(9) MCTS-transfer* (Wang et al., 2024), which employs Monte Carlo Tree Search (MCTS) to adaptively explore subspaces for improved design generation, and *(10) CMA-ES* (Hansen, 2006), an evolutionary baseline that models the design via a covariance matrix over the design space and samples candidates from there.

## 4.3. Implementation Details

We initialize our model from the pretrained diffusion language model LLADA-8B-INSTRUCT (Nie et al., 2025). The training process follows a sequential pipeline: domain adaptation (DA), supervised fine-tuning (SFT), and reinforcement learning (RL). For discrete tasks, we employ learning rates of $2 \times 10^{-5}$ for DA and SFT, and $1 \times 10^{-6}$

*Table 1.* **Main results on Design-Bench benchmarks.** We report the 100-th percentile normalized oracle score over 128 generated candidates on four tasks: Ant Morphology, D'Kitty Morphology, TF Bind 8, and TF Bind 10. Results are averaged over 8 random seeds (mean ± std). Rank Mean and Rank Median denote the average and median rank across all tasks. The highest and second-highest scores per task are highlighted in green and blue, respectively.

| Method | Ant Morphology | D'Kitty Morphology | TF Bind 8 | TF Bind 10 | Mean Score ↑ | Rank Mean ↓ | Rank Median ↓ |
|---|---|---|---|---|---|---|---|
| $\mathcal{D}$(**best**) | 0.565 | 0.884 | 0.439 | 0.511 | − | − | − |
| Grad-mean | $0.709 \pm 0.002$ | $0.920 \pm 0.008$ | $0.843 \pm 0.082$ | $0.736 \pm 0.016$ | $0.802 \pm 0.027$ | 4.25 | 3.5 |
| Grad-EI | $0.655 \pm 0.002$ | $0.923 \pm 0.010$ | $0.864 \pm 0.091$ | $0.727 \pm 0.024$ | $0.792 \pm 0.032$ | 4.25 | 4.0 |
| COMs | $0.647 \pm 0.020$ | $0.934 \pm 0.008$ | $0.843 \pm 0.046$ | $0.709 \pm 0.025$ | $0.783 \pm 0.025$ | 4.5 | 4.5 |
| ICT | $0.555 \pm 0.045$ | $0.932 \pm 0.037$ | $0.753 \pm 0.050$ | $0.585 \pm 0.014$ | $0.706 \pm 0.037$ | 10.0 | 11.5 |
| MATCH-OPT | $0.537 \pm 0.024$ | $0.925 \pm 0.025$ | $0.697 \pm 0.008$ | $0.583 \pm 0.034$ | $0.686 \pm 0.023$ | 11.75 | 12.5 |
| UniSO-T | $0.636 \pm 0.045$ | $0.939 \pm 0.007$ | $0.836 \pm 0.027$ | $0.522 \pm 0.017$ | $0.733 \pm 0.024$ | 6.75 | 6.5 |
| CbAS | $0.480 \pm 0.019$ | $0.911 \pm 0.035$ | $0.721 \pm 0.028$ | $0.597 \pm 0.005$ | $0.677 \pm 0.022$ | 13.0 | 13.5 |
| ExPT | $0.929 \pm 0.049$ | $\mathbf{0.950 \pm 0.041}$ | $0.810 \pm 0.044$ | $0.703 \pm 0.022$ | $0.848 \pm 0.039$ | 4.0 | 4.0 |
| MIN | $0.570 \pm 0.003$ | $0.886 \pm 0.017$ | $0.764 \pm 0.008$ | $0.517 \pm 0.030$ | $0.684 \pm 0.015$ | 12.25 | 12.5 |
| BONET | $0.632 \pm 0.042$ | $0.920 \pm 0.040$ | $0.776 \pm 0.007$ | $0.492 \pm 0.043$ | $0.705 \pm 0.033$ | 10.25 | 8.5 |
| OPRO | $0.517 \pm 0.039$ | $0.856 \pm 0.046$ | $0.758 \pm 0.017$ | $0.500 \pm 0.013$ | $0.658 \pm 0.029$ | 14.0 | 14.5 |
| GTG | $0.603 \pm 0.039$ | $0.917 \pm 0.023$ | $0.762 \pm 0.016$ | $0.730 \pm 0.026$ | $0.753 \pm 0.026$ | 8.25 | 9.5 |
| DDOM | $0.590 \pm 0.026$ | $0.929 \pm 0.037$ | $0.739 \pm 0.016$ | $0.497 \pm 0.002$ | $0.689 \pm 0.020$ | 11.25 | 12.5 |
| CMA-ES | $0.592 \pm 0.010$ | $0.711 \pm 0.045$ | $0.784 \pm 0.029$ | $0.658 \pm 0.031$ | $0.686 \pm 0.029$ | 10.25 | 9.0 |
| MCTS-transfer | $0.648 \pm 0.001$ | $0.910 \pm 0.006$ | $0.857 \pm 0.015$ | $0.628 \pm 0.043$ | $0.761 \pm 0.016$ | 7.25 | 6.5 |
| **DiBO**(ours) | $\mathbf{0.932 \pm 0.022}$ | $0.912 \pm 0.017$ | $\mathbf{0.946 \pm 0.043}$ | $\mathbf{0.741 \pm 0.027}$ | $\mathbf{0.883 \pm 0.027}$ | **3.5** | **1.0** |

for RL. For continuous tasks, the learning rates are $1 \times 10^{-5}$ (DA), $2 \times 10^{-5}$ (SFT), and $1 \times 10^{-6}$ (RL). Each stage begins with a 100-step linear warmup followed by a constant learning rate schedule. We use a per-device batch size of 1, with gradient accumulation steps set to 16 for discrete tasks and 8 for continuous tasks. For DA stage, we train the model for $1,024$ and $2,048$ optimization steps for discrete and continuous tasks respectively. For SFT and RL stages, we optimize all tasks with $1,024$ and 128 steps. Factoring in gradient accumulation, this equates to processing $16,384$ / $16,384$ / $2,048$ total samples for discrete tasks, and $16,384$ / $8,192$ / $1,024$ samples for continuous tasks. We adopt the `PagedAdamW8bit` optimizer as implemented in the `bitsandbytes` library, building on block-wise quantized optimizers (Dettmers et al., 2022), with Bfloat16 precision. All diffusion-related hyperparameters, including the masking schedule and mask ratio, remain consistent with the official implementation by Nie et al. (2025).

For each task, we construct a fixed offline pool of $n_{\text{pool}} = 500$ samples by evenly sub-sampling the label-sorted dataset. To remain within the memory constraints of a single NVIDIA H100 (80 GB) GPU, we use $n_{\text{few}} = 7$ few-shot examples per prompt. At inference time, for each candidate, we independently resample the corresponding context examples from the offline pool. We complete the masked response in a single forward pass using greedy token filling without temperature-based stochastic decoding. Duplicate outputs are discarded, and generation continues until 128 unique valid candidates are obtained. Detailed sampling rules, text rendering formats, and prompt templates are provided in Appendix A.1, A.2, and A.3. All experiments are performed on a single H100 GPU; for instance, the entire

pipeline for the TF Bind 8 task can be completed within 1.5 hours. All results are reported as mean ± standard deviation, averaged over 8 different random seeds.

### 4.4. Results and Analysis

Following previous practice (Yuan et al., 2023; Nguyen et al., 2023), Table 1 reports the normalized 100-th percentile oracle scores on four Design-Bench tasks. Our method, **DiBO**, achieves the best overall performance, ranking first in terms of mean rank and median rank across all compared methods. In addition, we report median (50-th percentile) results in Appendix A.5, where DiBO also achieves state-of-the-art performance, highlighting the overall effectiveness of our approach.

Across tasks, DiBO consistently outperforms the strongest baseline on three out of four benchmarks—Ant Morphology, TF Bind 8, and TF Bind 10. Notably, on TF Bind 8, DiBO surpasses the best baseline by 0.082 in normalized score, representing a substantial improvement. On D'Kitty Morphology, DiBO underperforms the best baseline by 0.038, while remaining competitive with other strong methods. Overall, these results demonstrate the strong effectiveness of DiBO across diverse tasks and design domains.

When compared to *forward methods*, which rely on learned surrogate models and gradient-based optimization, DiBO consistently demonstrates superior performance. An explanation is that forward methods are inherently sensitive to surrogate modeling errors and distributional shift: surrogate models trained on offline data may produce inaccurate gradients when optimizing toward regions that are poorly supported by the dataset. In contrast, DiBO directly models

*Table 2.* **Matched comparison between diffusion and autoregressive (AR) backbones.** We construct an autoregressive counterpart (LLaMA-3.1-8B-Instruct) by keeping the prompt construction, dataset, delimiter tokens, context selection strategy, and the full post-training pipeline (DA→SFT→RL) unchanged, and replacing only the backbone architecture (diffusion vs. AR). Results report the 100-th percentile normalized oracle score (mean ± std over seeds). The last row ("Performance Gain") shows the absolute improvement of the diffusion backbone over the matched AR backbone at each stage.

*(a)* Discrete Tasks

| Method | TF Bind 8 | | | TF Bind 10 | | |
|---|---|---|---|---|---|---|
| | DA | SFT | RL | DA | SFT | RL |
| Autoregressive | $0.803 \pm 0.038$ | $0.875 \pm 0.008$ | $0.915 \pm 0.008$ | $0.623 \pm 0.047$ | $0.633 \pm 0.036$ | $0.682 \pm 0.053$ |
| DiBO$_{(ours)}$ | $0.883 \pm 0.032$ | $0.939 \pm 0.031$ | $0.946 \pm 0.043$ | $0.644 \pm 0.039$ | $0.704 \pm 0.056$ | $0.741 \pm 0.027$ |
| Performance Gain | 0.080 ↑↑ | 0.064 ↑↑ | 0.031 ↑ | 0.021 ↑ | 0.071 ↑↑ | 0.059 ↑↑ |

*(b)* Continuous Tasks

| Method | Ant Morphology | | | D'Kitty Morphology | | |
|---|---|---|---|---|---|---|
| | DA | SFT | RL | DA | SFT | RL |
| Autoregressive | $0.857 \pm 0.044$ | $0.894 \pm 0.032$ | $0.930 \pm 0.000$ | $0.888 \pm 0.025$ | $0.905 \pm 0.005$ | $0.912 \pm 0.011$ |
| DiBO$_{(ours)}$ | $0.875 \pm 0.026$ | $0.929 \pm 0.017$ | $0.932 \pm 0.022$ | $0.903 \pm 0.014$ | $0.908 \pm 0.009$ | $0.912 \pm 0.017$ |
| Performance Gain | 0.018 ↑ | 0.035 ↑ | 0.002 ↑ | 0.015 ↑ | 0.003 ↑ | 0.000 |

the design distribution through generative denoising and avoids explicit reliance on surrogate gradients, resulting in more robust optimization under limited offline data.

Relative to *inverse methods*, including VAE-, GAN-, autoregressive-, and diffusion-based approaches, DiBO also achieves strong and often superior performance. While existing LLM-based optimization methods have shown encouraging performance, they adopt different modeling assumptions that limit their applicability. AR LLM approaches such as OPRO (Yang et al., 2024) generate designs sequentially from left to right, which can make it challenging to capture bidirectional dependencies. Diffusion-based inverse methods such as DDOM (Krishnamoorthy et al., 2023) typically operate in continuous design spaces with task-specific architectures, and do not natively support the integration of textual task descriptions. DiBO instead performs optimization directly in a unified discrete token space using diffusion language modeling, enabling bidirectional context modeling and seamless integration of textual instructions, designs, and labels. This design choice contributes to DiBO's robust and consistent performance across diverse tasks.

Beyond the main results, Appendix A.6 reports Top-$K$ metrics and three additional RNA design tasks. Furthermore, we audit whether the pretrained LLADA-8B-INSTRUCT model already contains task-specific design-label knowledge from public benchmark assets, as detailed in Appendix A.7. Overall, these results demonstrate that DiBO provides a robust and general framework for offline black-box optimization, achieving strong performance across both discrete sequence and continuous domains while maintaining favorable aggregate rankings across tasks.

**Comparison with Inference-Only Diffusion Optimization** Yuan et al. (2026) also explore diffusion LLMs for black-box optimization, but focus on an inference-only setting that queries a frozen diffusion LLM with MCTS guidance and does not update model parameters. In contrast, DiBO adapts the diffusion LLM to offline BBO through domain adaptation, supervised fine-tuning, and reinforcement learning, enabling the model to internalize BBO-specific prompt formats, design-label structures, and reward feedback. Moreover, their method targets an extremely few-shot regime (e.g., $n_{pool} = 10$), whereas our experiments use a small-data offline setting with $n_{pool} = 500$. We therefore exclude their method from our main baselines and view it as complementary to our training-based framework.

### 4.5. Ablation Studies

We ablate the key components of our training pipeline to assess their individual contributions, conducted under the same evaluation protocol as the main experiments.

**Matched Diffusion vs. Autoregressive Modeling** One of our central motivations is that many design problems exhibit bidirectional dependencies, where each design variable can be influenced by both its prefix and suffix. To this end, we conduct a matched comparison between diffusion and autoregressive (AR) backbones, by replacing LLADA-8B-INSTRUCT with LLAMA-3.1-8B-INSTRUCT while keeping all other settings unchanged.

As shown in Table 2, the diffusion backbone outperforms or matches the corresponding AR backbone across all tasks and training stages. The gains are especially pronounced on

*Table 3.* **Ablation of the usage of delimiter tokens.** Plain text (e.g., "Designs:" and "Labels:") is used to mark boundaries when delimiter tokens are absent. We report performance after sequential training up to each stage (DA, SFT, and RL) on TF Bind 8 and Ant Morphology, and additionally report the per-stage performance gain brought by using delimiter tokens.

| Method | TF Bind 8 | | | Ant Morphology | | |
|---|---|---|---|---|---|---|
| | DA | SFT | RL | DA | SFT | RL |
| w/o delimiter tokens | $0.859 \pm 0.038$ | $0.918 \pm 0.043$ | $0.943 \pm 0.035$ | $0.693 \pm 0.016$ | $0.903 \pm 0.022$ | $0.923 \pm 0.019$ |
| w/ delimiter tokens$_{(ours)}$ | $0.883 \pm 0.032$ | $0.939 \pm 0.031$ | $0.946 \pm 0.043$ | $0.875 \pm 0.026$ | $0.929 \pm 0.017$ | $0.932 \pm 0.022$ |
| Performance Gain | $0.024 \uparrow$ | $0.021 \uparrow$ | $0.003 \uparrow$ | $0.182 \uparrow\uparrow$ | $0.026 \uparrow$ | $0.009 \uparrow$ |

*Table 4.* **Ablation of training stages.** We remove certain training stages from the three-stage pipeline. Performance drop $\Delta$ is measured relative to the full pipeline (DA+SFT+RL). "✗" indicates no valid output is produced due to formatting issues.

| Stages | | | TF Bind 8 | | Ant Morphology | |
|---|---|---|---|---|---|---|
| DA | SFT | RL | Score | $\Delta$ | Score | $\Delta$ |
| ✓ | | | $0.883 \pm 0.032$ | $-0.063 \downarrow$ | $0.875 \pm 0.026$ | $-0.057 \downarrow$ |
| | ✓ | | $0.938 \pm 0.034$ | $-0.008 \downarrow$ | $0.898 \pm 0.027$ | $-0.034 \downarrow$ |
| | | ✓ | $0.437 \pm 0.001$ | $-0.509 \downarrow\downarrow$ | ✗ | – |
| ✓ | ✓ | | $0.939 \pm 0.031$ | $-0.007 \downarrow$ | $0.929 \pm 0.017$ | $-0.003 \downarrow$ |
| ✓ | | ✓ | $0.928 \pm 0.052$ | $-0.018 \downarrow$ | $0.858 \pm 0.032$ | $-0.074 \downarrow$ |
| | ✓ | ✓ | $0.941 \pm 0.030$ | $-0.005 \downarrow$ | $0.913 \pm 0.024$ | $-0.019 \downarrow$ |
| ✓ | ✓ | ✓ | $\mathbf{0.946 \pm 0.043}$ | – | $\mathbf{0.932 \pm 0.022}$ | – |

*Table 5.* **Ablation on similarity-conditioned context construction.** We compare selecting prompt examples based on design similarity against removing this locality constraint and using randomly selected context examples.

| Setting | TF Bind 8 | Ant Morphology |
|---|---|---|
| Random Context | $0.730 \pm 0.001$ | $0.906 \pm 0.008$ |
| Similar Context$_{(ours)}$ | $0.946 \pm 0.043$ | $0.932 \pm 0.022$ |
| Performance Gain | $0.216 \uparrow\uparrow$ | $0.026 \uparrow$ |

**Training Stages** We evaluate variants that remove one stage from the three-stage pipeline (*DA+SFT+RL*), resulting in *DA+SFT*, *SFT+RL*, and *DA+RL*. As shown in Table 4, across both tasks, removing any stage leads to a clear performance drop compared to the full pipeline, highlighting the importance of all three stages.

Among the two-stage variants, *DA+RL* exhibits the largest degradation in performance, indicating that reinforcement learning without an intermediate supervised stage is insufficient to reliably exploit fine-grained prompt–response mappings, especially in the continuous domain. Similarly, *SFT+RL* underperforms the full pipeline, suggesting that skipping domain adaptation leads to suboptimal learning when handling heterogeneous prompt components. *DA+SFT* also exhibits a consistent gap, highlighting that our full pipeline benefits from incorporating reinforcement learning as a fine-grained optimization stage on top of a well-aligned representation space.

All single-stage variants perform substantially worse than the integrated approach. In particular, *RL-only* fails to produce valid outputs on Ant Morphology, highlighting that RL without prior alignment struggles to correctly learn the heterogeneous components in complex design tasks. Overall, these results confirm that the three training stages are complementary and jointly necessary to achieve strong and consistent performance across tasks. In particular, these results emphasize the importance of a well-structured training pipeline, where different stages play complementary roles rather than acting in isolation.

the discrete DNA sequence tasks, where diffusion improves over AR by up to $0.080$ on TF Bind 8 and $0.071$ on TF Bind 10. This supports our hypothesis that bidirectional refinement is particularly beneficial for structured design problems, where functional dependencies are often non-sequential. The gains are smaller but still positive on the continuous tasks, suggesting that the advantage of diffusion modeling also extends beyond discrete sequence generation.

**Delimiter Tokens** We remove the design and label delimiter tokens (1) `|design-start|`/`|design-end|` and (2) `|label-start|`/`|label-end|` and train the model using plain text inputs (e.g., "Designs: " and "Labels: "). Performance is evaluated after sequential training up to each stage as shown in Table 3.

Removing these delimiters leads to consistent performance degradation across all three training stages (DA, SFT, and RL) on both tasks. The performance drop is particularly pronounced during the domain adaptation stage on Ant Morphology, which further underscores the importance of delimiter tokens in enabling effective domain adaptation, especially for complex continuous spaces.

Overall, these results demonstrate that explicit delimiter tokens play a critical role throughout the entire training pipeline, by providing a clear separation between heterogeneous input components and facilitating effective learning across different stages.

**Similarity-Conditioned Context Construction** We study the effect of conditioning the target response to remain within a reasonable distribution of the prompt examples. As shown in Table 5, this similarity-conditioned context is

*Table 6.* **Comparison between evenly spaced sub-sampling and random sub-sampling of the offline dataset.** We report the 100th-percentile normalized oracle score on four tasks, as well as the performance difference between the two strategies.

| Method | Ant Morphology | D'Kitty Morphology | TF Bind 8 | TF Bind 10 |
|---|---|---|---|---|
| Random sub-sampling | $0.936 \pm 0.019$ | $0.909 \pm 0.008$ | $0.960 \pm 0.027$ | $0.743 \pm 0.012$ |
| Evenly sub-sampling$_{(ours)}$ | $0.932 \pm 0.022$ | $0.912 \pm 0.017$ | $0.946 \pm 0.043$ | $0.741 \pm 0.027$ |
| Difference | $-0.004$ | $+0.003$ | $-0.014$ | $-0.002$ |

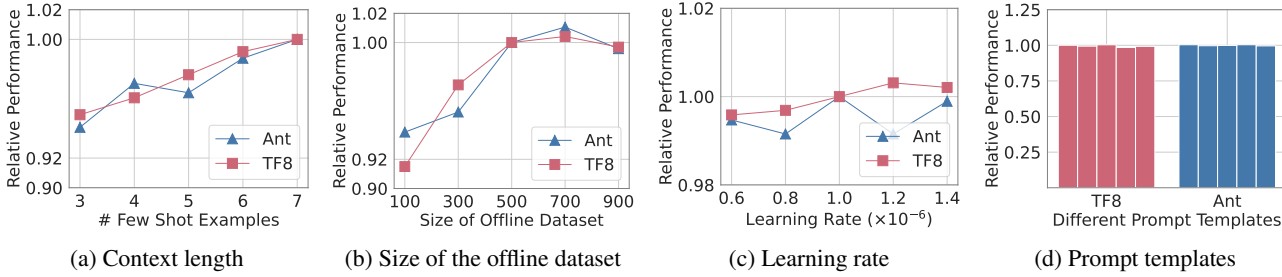

| (a) Context length | (b) Size of the offline dataset | (c) Learning rate | (d) Prompt templates |
|---|---|---|---|

*Figure 2.* **Hyperparameter sensitivity** at the RL stage. Results are reported as relative performance normalized by our default setting.

critical for performance. Removing this constraint leads to a substantial degradation on both tasks, with a particularly severe drop on TF Bind 8. These results indicate that diffusion LLMs benefit from prompt contexts that preserve local semantic consistency, which facilitates effective reasoning over the underlying design landscape during generation.

**Evenly Spaced Sub-Sampling**  We examine the effect of evenly sub-sampling the label-sorted dataset, which provides reproducible coverage of the label spectrum. As shown in Table 6, evenly spaced and random sub-sampling achieve similar performance across all four tasks, with differences around $0.01$ normalized score. Thus, our results are not sensitive to this offline-pool sampling strategy.

### 4.6. Hyperparameter Sensitivity

We evaluate the sensitivity of our method to several critical hyperparameters. Unless otherwise specified, all experiments are conducted using the primary configuration detailed in Section 4.3.

**Context Length**  We investigate the impact of the number of few-shot examples in the prompt, varying the count from 3 to 7 (see Figure 2(a)). Performance demonstrates a steady upward trend as more examples are included. However, due to GPU memory constraints, we capped the context length at 7 examples. Consequently, we utilize 7 few-shot examples for all main experiments to maximize in-context information while remaining within hardware limits.

**Size of Offline Dataset**  We vary the offline pool size $n_{pool}$ within $\{100, 300, 500, 700, 900\}$, where the default size is $500$. As shown in Figure 2(b), performance is worst when only 100 samples are used, and improves consistently as the pool size increases on both tasks. A slight drop when $n_{pool}$ reaches 900 is observed.

**Learning Rate of Reinforcement Learning**  To assess the stability of the Reinforcement Learning (RL) phase, we vary the learning rate among $\{0.6, 0.8, 1.0, 1.2, 1.4\} \times 10^{-6}$ and report the results in Figure 2(c). We observe that the model's performance remains largely invariant across this range, suggesting that our RL training objective is robust and relatively insensitive to specific learning rate tuning.

**Prompt Template Variation**  We evaluate sensitivity to natural prompt paraphrases by using five sets of prompt templates generated by ChatGPT-5.2 and re-training the model on TF Bind 8 and Ant Morphology. These templates preserve the same task semantics, delimiter tokens, and output format, but vary the surface-level instruction wording. As shown in Figure 2(d), performance remains stable across these paraphrastic variants. These results indicate that DiBO is stable under natural paraphrases of the task instruction, as long as the task semantics and structured output format are preserved. More details on the prompt examples we use are provided in Appendix A.3.

### 5. Conclusion

We present DiBO, a framework for adapting diffusion language models to black-box optimization. To bridge the semantic gap between natural language and offline design-label data, we introduce explicit delimiter tokens and a joint prompt–response reconstruction loss for domain adaptation. We further combine supervised fine-tuning and reinforcement learning in a two-stage post-training pipeline, enabling the model to synthesize high-label designs from complex, heterogeneous inputs. Experiments across diverse discrete and continuous Design-Bench tasks show strong and consistent performance, underscoring the promise of diffusion language models as a robust, general-purpose paradigm for offline optimization in high-dimensional design spaces.

## Impact Statement

This paper presents a diffusion-based language model framework for offline black-box optimization, aiming to accelerate discovery in scientific and engineering domains such as molecular sequence design, robotics, and materials science. By reducing reliance on resource-intensive physical experiments and avoiding online trial-and-error, our approach offers significant advantages in cost-efficiency and safety for sensitive applications.

However, as a general-purpose generative optimization method, it carries potential risks if applied without appropriate safeguards. The model may inherit or amplify biases present in the offline training datasets. Furthermore, while intended to assist scientific discovery, we acknowledge the risk of misuse in generating harmful designs (e.g., in biochemical contexts). We emphasize that this tool is designed to augment expert judgment, not replace it, and its deployment requires responsible oversight.

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

# A. Appendix

## A.1. Offline Data Construction

For each Design-Bench task, we load the original designs $\{x_i^{\text{raw}}\}$ and their corresponding scalar labels $\{y_i\}$. Here, $x_i^{\text{raw}}$ denotes the task-specific raw representation of a design (e.g., discrete sequences for TF Bind 8 / 10 tasks or continuous vectors for robot design tasks). All samples are sorted by label value, and an evenly spaced subset of size $n_{\text{pool}}$ is selected to form the offline dataset. Unless otherwise stated, the offline pool size $n_{\text{pool}}$ is set to $500$ in all main experiments. This pool remains fixed and is used across all training stages of our method as well as for all comparison methods.

## A.2. Prompt-Response Construction

**Partitioning Strategy** We partition the offline data into two disjoint subsets, $D_1$ and $D_2$, with a fixed ratio of $0.8 : 0.2$. $D_1$ corresponds to the sampling pool for prompt context (designs with lower scores), while $D_2$ corresponds to the sampling pool for response targets (designs with higher scores). We strictly separate $D_1$ and $D_2$ to prevent data leakage; specifically, we ensure that no design used as a target response appears in the few-shot prompt context.

**Design Similarity** Forming prompt-response pairs by randomly sampling from $D_1$ and $D_2$ is often suboptimal. If the designs in the prompt context are structurally dissimilar to the target design in the response, the model faces a significant challenge in learning the underlying local optimization landscape. To address this, we construct prompt-response pairs by selecting designs from the prompt context that are most similar to the target response. Furthermore, for the RL phase, we balance the offline dataset to include an approximately equal distribution of positive and negative reward signals (i.e., improvements vs. degradations relative to the anchor).

We quantify the similarity between designs using a normalized representation, $x^{\text{norm}}$. For sequence-based tasks (e.g., TF Bind 8 / 10), $x^{\text{norm}}$ is obtained by applying `map_to_logits()` to the discrete sequence, followed by feature-wise normalization. For continuous robot tasks, we directly utilize the normalized continuous features provided by Design-Bench. The similarity between two designs $i$ and $j$ is defined using an RBF kernel:

$$\text{sim}(i, j) = \exp\left(-\frac{\|x_i^{\text{norm}} - x_j^{\text{norm}}\|_2^2}{2\sigma^2}\right),$$

where the kernel bandwidth $\sigma$ is the median of all pairwise Euclidean distances computed across the entire offline pool.

## A.3. Prompt Templates

We use a total of 10 natural-language prompt templates to introduce linguistic diversity: 8 templates are used for training, and 2 templates are held out for validation. All templates preserve the same optimization objective, delimiter tokens, design-label structure, and output format, and differ only in surface-level natural-language wording. The paraphrases are generated by ChatGPT-5.2.

*Table 7.* **Representative prompt paraphrases for TF Bind** 8. All variants preserve the same task semantics, delimiter tokens, and output format, while varying only the surface-level instruction wording.

| Template | Instruction wording |
|---|---|
| T1 | You are a helpful optimization assistant that will help us generate a new length-8 optimal DNA sequence with maximum binding affinity with a particular transcription factor `SIX6_REF_R1`. |
| T2 | The goal is to design an 8-length DNA sequence with maximum binding affinity to the transcription factor `SIX6_REF_R1`. |
| T3 | Optimization Task: Find a DNA sequence (length = 8) that binds more strongly to `SIX6_REF_R1`. Training data (sequence, affinity) is as follows. |
| T4 | You are assisting a molecular biologist in optimizing DNA binding sites. |
| T5 | Below is a reference dataset of DNA sequences ranked by binding affinity to the transcription factor `SIX6_REF_R1`. |

**Paraphrased Template Examples** Table 7 shows representative prompt-instruction variants for TF Bind 8. All variants describe the same optimization goal (i.e., designing a length-8 DNA sequence with improved binding affinity to the same transcription factor), but use different surface-level wording. For example, some variants state the task as a direct goal, while others introduce it as a reference dataset or an optimization problem. After this instruction part, all variants use the same

structured design-label context with `|design-start|`, `|design-end|`, `|label-start|`, and `|label-end|`, and require the response to follow the same delimiter-based design format.

**Example Prompt Instantiations** We further provide one full prompt instantiation for each task as better reference. These examples illustrate how the task description, few-shot design-label context, delimiter tokens, and response format are combined in the actual inputs. For continuous tasks, vector entries are abbreviated with ellipses for readability, while the actual inputs contain the full design vectors.

---

**TF Bind 8 (Discrete sequence)**

You are a helpful optimization assistant tasked with generating a length-8 DNA sequence that maximizes binding affinity with a specific transcription factor (`SIX6_REF_R1`).
You are given several existing DNA sequences along with their corresponding binding affinities:
`|design-start|['A','C','G','T','A','C','G','T']|design-end|`
`|label-start|+000.018|label-end|`
`|design-start|['T','C','G','C','T','A','G','G']|design-end|`
`|label-start|+000.027|label-end|`
Please propose a new DNA sequence that is different from the existing sequences and has a higher binding affinity.
Response: `|design-start|['G','C','G','T','A','C','G','T']|design-end|`

---

**TF Bind 10 (Discrete sequence)**

You are a helpful optimization assistant tasked with generating a length-10 DNA sequence that optimizes binding to a specific transcription factor (`Pho4`).
Goal: maximize $-\mathrm{ddG}$ (larger values indicate stronger binding).
You are given several existing DNA sequences along with their corresponding $-\mathrm{ddG}$ values:
`|design-start|['T','C','C','A','C','G','A','A','G','A']|design-end|`
`|label-start|-001.860|label-end|`
`|design-start|['G','C','T','T','G','G','A','A','C','A']|design-end|`
`|label-start|+000.010|label-end|`
Please propose a new DNA sequence that is different from the existing sequences and achieves a higher $-\mathrm{ddG}$ value than all sequences shown above. The DNA sequence should use only A, C, G, and T, and it should differ from each existing sequence in at least one position.
Response: `|design-start|['A','C','G','T','A','C','G','T','A','C']|design-end|`

---

**AntMorphology (Continuous control)**

You are a helpful optimization assistant for Ant robot morphology design in OpenAI Gym. The goal is to maximize the running performance of a simulated quadruped robot (Ant). Each morphology is represented by 60 continuous parameters (4 legs × 15 parameters per leg).
For each leg, parameters are ordered as follows: $x$ (hip x), $y$ (hip y), $z$ (hip z), $a$ (hip angle), $b$ (thigh angle), $c$ (ankle angle), hip center, hip range, thigh center, thigh range, ankle center, ankle range, hip size, thigh size, ankle size.
Existing designs and their performance scores:
`|design-start|[+001.000, +010.000, ..., -002.345, +000.091]|design-end|`
`|label-start|+000.742|label-end|`
`|design-start|[+002.500, +008.750, ..., -001.120, +000.305]|design-end|`
`|label-start|+000.768|label-end|`
Please propose one new morphology design that is different from the above designs and is expected to achieve a higher performance score. All parameters must be floating-point numbers rounded to three decimal places.
Response:
`|design-start|[+002.000, +009.500, ..., -001.800, +000.250]|design-end|`

---

---

**D'KittyMorphology (Continuous control)**

You are a helpful optimization assistant for D'Kitty robot morphology design in simulation.
Goal: maximize the performance score (better locomotion/navigation to a fixed target; higher is better).
Each design is 56 continuous parameters (4 legs × 14 parameters per leg).
Per-leg parameters (in order): $x$ (hip x), $y$ (hip y), $z$ (hip z), $a$ (hip angle), $b$ (knee angle), hip center, hip range, knee center, knee range, hip size, knee size, foot center, foot range, foot size.
Existing designs and their performance scores:
```
|design-start|[+001.250, +009.000, ..., -001.400, +000.180]|design-end|
|label-start|+000.703|label-end|
|design-start|[+002.100, +008.500, ..., -000.950, +000.260]|design-end|
|label-start|+000.729|label-end|
```
Please propose one new morphology design to maximize the score.
Format: each value must be a signed float with three decimal places (e.g., +000.123, -001.500).
Response:
```
|design-start|[+001.800, +009.250, ..., -001.200, +000.220]|design-end|
```

---

## A.4. Effect of High-Dimensional GP Priors

*Table 8.* **Effect of high-dimensional GP kernels on baseline performance.** We compare a plain RBF GP with the high-dimensional formulation of Hvarfner et al. (2024), which incorporates dimensionality-scaled priors (DSP) for stable modeling in high-dimensional settings. We report both the maximum and median scores, and use $\Delta$ to denote the change of performance from plain RBF to the high-dimensional configuration.

*(a)* Maximum Scores

| Method | Kernel | Ant Morphology | D'Kitty Morphology | TF Bind 8 | TF Bind 10 |
|--------|--------|----------------|--------------------|-----------|------------|
| Grad-mean | RBF | $0.644 \pm 0.039$ | $0.907 \pm 0.016$ | $0.666 \pm 0.011$ | $0.695 \pm 0.027$ |
| | High-dim | $0.709 \pm 0.002$ | $0.920 \pm 0.008$ | $0.843 \pm 0.082$ | $0.736 \pm 0.016$ |
| | $\Delta$ | $+0.065 \uparrow$ | $+0.013 \uparrow$ | $+0.177 \uparrow$ | $+0.041 \uparrow$ |
| Grad-EI | RBF | $0.626 \pm 0.002$ | $0.901 \pm 0.045$ | $0.673 \pm 0.012$ | $0.689 \pm 0.013$ |
| | High-dim | $0.655 \pm 0.002$ | $0.923 \pm 0.010$ | $0.864 \pm 0.091$ | $0.727 \pm 0.024$ |
| | $\Delta$ | $+0.029 \uparrow$ | $+0.022 \uparrow$ | $+0.191 \uparrow$ | $+0.038 \uparrow$ |
| **DiBO**(ours) | – | $0.932 \pm 0.022$ | $0.912 \pm 0.017$ | $0.946 \pm 0.043$ | $0.741 \pm 0.027$ |

*(b)* Median Scores

| Method | Kernel | Ant Morphology | D'Kitty Morphology | TF Bind 8 | TF Bind 10 |
|--------|--------|----------------|--------------------|-----------|------------|
| Grad-mean | RBF | $0.351 \pm 0.002$ | $0.879 \pm 0.015$ | $0.623 \pm 0.030$ | $0.394 \pm 0.004$ |
| | High-dim | $0.472 \pm 0.013$ | $0.858 \pm 0.003$ | $0.441 \pm 0.008$ | $0.461 \pm 0.008$ |
| | $\Delta$ | $+0.121 \uparrow$ | $-0.021 \downarrow$ | $-0.182 \downarrow$ | $+0.067 \uparrow$ |
| Grad-EI | RBF | $0.368 \pm 0.046$ | $0.872 \pm 0.004$ | $0.595 \pm 0.019$ | $0.453 \pm 0.047$ |
| | High-dim | $0.504 \pm 0.013$ | $0.849 \pm 0.008$ | $0.436 \pm 0.004$ | $0.459 \pm 0.024$ |
| | $\Delta$ | $+0.136 \uparrow$ | $-0.023 \downarrow$ | $-0.159 \downarrow$ | $+0.006 \uparrow$ |
| **DiBO**(ours) | – | $0.524 \pm 0.038$ | $0.775 \pm 0.010$ | $0.473 \pm 0.020$ | $0.491 \pm 0.006$ |

In Bayesian optimization, GP surrogates provide a standard way to model unknown black-box objectives and guide acquisition-based search (Garnett, 2023). However, in high-dimensional settings, the effectiveness of GP-based BO can depend strongly on kernel choices and hyperparameter priors, especially the lengthscale prior (Binois & Wycoff, 2022). Given that such high-dimensional challenges are increasingly prevalent in recent BBO (Chen et al., 2023a;b; Yuan et al., 2024; 2025; Yang et al., 2026) and multi-modal studies (Lee et al., 2024; Zhang et al., 2026b;a; Du et al., 2026; Yan et al., 2026), addressing this scaling issue is critical. Fortunately, recent work shows that a simple dimension-scaled GP prior can

substantially improve vanilla BO in high-dimensional search spaces (Hvarfner et al., 2024). Following this line of work, we implement our GP-based baselines following BoTorch (Balandat et al., 2020) and adopt the dimension-scaled GP prior configuration of Hvarfner et al. (2024) for both *Grad (mean)* and *Grad (EI)*. Readers can refer to a detailed discussion here. This configuration scales the GP lengthscale prior with the input dimensionality, making the surrogate better calibrated for high-dimensional design spaces. All other components of the baselines are kept fixed.

To justify this choice, we further compare the adopted high-dimensional GP configuration with a plain RBF GP configuration in Table 8. The high-dimensional configuration consistently improves the GP-based baselines in most maximum-score comparisons and in several median-score comparisons, empirically confirming that GP prior calibration is an important implementation detail in high-dimensional design spaces. These results also show that our main conclusions remain robust under a stronger and more carefully calibrated GP-based baseline configuration.

## A.5. Median Results

This section reports the median (50-th percentile) normalized oracle scores on all Design-Bench tasks. Consistent with the main results in Table 1, DiBO achieves state-of-the-art or competitive median performance across tasks.

*Table 9.* **Experimental results in** $50$**-th percentile normalized scores on four tasks for comparison.** Results are averaged over 8 random seeds (mean ± std). Rank Mean and Rank Median denote the average and median rank across all tasks, respectively. The highest and second-highest scores per task are highlighted in **green** and blue, respectively.

| Method | Ant Morphology | D'Kitty Morphology | TF Bind 8 | TF Bind 10 | Mean Score ↑ | Rank Mean ↓ | Rank Median ↓ |
|---|---|---|---|---|---|---|---|
| $\mathcal{D}$(**best**) | 0.565 | 0.884 | 0.439 | 0.511 | − | − | − |
| Grad-mean | $0.472 \pm 0.013$ | $0.858 \pm 0.003$ | $0.441 \pm 0.008$ | $0.461 \pm 0.008$ | $0.558 \pm 0.008$ | **3.5** | 3.5 |
| Grad-EI | $0.504 \pm 0.013$ | $0.849 \pm 0.008$ | $0.436 \pm 0.004$ | $0.459 \pm 0.024$ | $0.562 \pm 0.012$ | 4.0 | 4.0 |
| COMs | $0.375 \pm 0.046$ | $\mathbf{0.887 \pm 0.007}$ | $0.399 \pm 0.048$ | $0.487 \pm 0.039$ | $0.537 \pm 0.035$ | 5.25 | 5.0 |
| ICT | $0.407 \pm 0.022$ | $0.863 \pm 0.049$ | $0.366 \pm 0.010$ | $0.365 \pm 0.017$ | $0.500 \pm 0.025$ | 9.25 | 9.0 |
| MATCH-OPT | $0.435 \pm 0.049$ | $0.875 \pm 0.022$ | $0.431 \pm 0.034$ | $0.373 \pm 0.007$ | $0.529 \pm 0.028$ | 6.25 | 4.5 |
| UniSO-T | $0.405 \pm 0.026$ | $0.842 \pm 0.043$ | $0.418 \pm 0.013$ | $0.431 \pm 0.015$ | $0.524 \pm 0.024$ | 7.25 | 7.0 |
| CbAS | $0.302 \pm 0.033$ | $0.773 \pm 0.047$ | $0.394 \pm 0.009$ | $0.373 \pm 0.009$ | $0.461 \pm 0.025$ | 13.0 | 13.5 |
| ExPT | $0.465 \pm 0.039$ | $0.792 \pm 0.016$ | $0.355 \pm 0.011$ | $0.356 \pm 0.027$ | $0.492 \pm 0.023$ | 11.5 | 13.0 |
| MIN | $0.370 \pm 0.002$ | $0.841 \pm 0.045$ | $0.282 \pm 0.012$ | $0.386 \pm 0.013$ | $0.470 \pm 0.018$ | 11.75 | 11.5 |
| BONET | $0.391 \pm 0.020$ | $0.885 \pm 0.008$ | $0.408 \pm 0.046$ | $0.474 \pm 0.025$ | $0.540 \pm 0.025$ | 4.75 | 4.5 |
| OPRO | $0.380 \pm 0.045$ | $0.829 \pm 0.037$ | $0.407 \pm 0.050$ | $0.432 \pm 0.014$ | $0.512 \pm 0.037$ | 9.0 | 9.0 |
| GTG | $0.372 \pm 0.024$ | $0.765 \pm 0.025$ | $0.360 \pm 0.008$ | $0.382 \pm 0.034$ | $0.470 \pm 0.023$ | 12.75 | 12.5 |
| DDOM | $0.323 \pm 0.045$ | $0.835 \pm 0.007$ | $0.373 \pm 0.027$ | $0.454 \pm 0.017$ | $0.496 \pm 0.024$ | 10.0 | 10.0 |
| CMA-ES | $0.335 \pm 0.019$ | $0.686 \pm 0.035$ | $0.388 \pm 0.028$ | $0.436 \pm 0.005$ | $0.461 \pm 0.022$ | 11.75 | 11.5 |
| MCTS-transfer | $0.323 \pm 0.049$ | $0.832 \pm 0.041$ | $0.310 \pm 0.044$ | $0.453 \pm 0.022$ | $0.480 \pm 0.039$ | 11.5 | 12.0 |
| **DiBO**$_{(ours)}$ | $\mathbf{0.524 \pm 0.038}$ | $0.775 \pm 0.010$ | $\mathbf{0.473 \pm 0.020}$ | $\mathbf{0.491 \pm 0.006}$ | $\mathbf{0.566 \pm 0.019}$ | 4.0 | **1.0** |

## A.6. Additional Evaluation on Top-$K$ Metrics and RNA Design Tasks

We provide additional evaluation along two dimensions. First, beyond the 100-th percentile score used in the main results, we report Top-$K$ average scores, where Top-$K$ denotes the average oracle score among the top $K$ generated candidates out of 128 candidates. We report Top-5, Top-10, Top-20, and the 50-th percentile score to characterize not only the best generated candidate but also the quality of the generated candidate set.

As shown in Table 10, DiBO remains strong on the original Design-Bench tasks under Top-$K$ and median metrics. In particular, it achieves the best scores across all reported metrics on Ant Morphology, TF Bind 8, and TF Bind 10, while remaining competitive on D'Kitty Morphology as discussed.

Second, we extend the evaluation to three RNA optimization tasks, RNA-A/B/C, adapted from BootGen (Kim et al., 2023). These tasks require designing length-14 RNA sequences to maximize transcription factor binding activity, providing an additional biological sequence domain beyond the Design-Bench tasks. We compare DiBO with COMs and ExPT, the two representative baselines from our main evaluation. The results are comparable but more mixed on the RNA tasks, as shown in Table 11. DiBO achieves the best 100-th percentile score on RNA-A and RNA-C, indicating that it can still discover high-scoring candidates in these additional biological sequence tasks.

*Table 10.* **Additional evaluation on Top-$K$ metrics on Design-Bench tasks.** We report the 100-th percentile, Top-5/10/20 average, and 50-th percentile normalized oracle scores over 128 generated candidates. The highest and second-highest scores per task are highlighted in **green** and blue , respectively.

| Task | Metric | COMs | ExPT | **DiBO**$_{(ours)}$ |
|---|---|---|---|---|
| Ant Morphology | 100-th | $0.647 \pm 0.020$ | $0.929 \pm 0.049$ | $\mathbf{0.932 \pm 0.022}$ |
| | Top-5 | $0.628 \pm 0.014$ | $0.904 \pm 0.028$ | $\mathbf{0.906 \pm 0.012}$ |
| | Top-10 | $0.621 \pm 0.020$ | $0.863 \pm 0.041$ | $\mathbf{0.880 \pm 0.010}$ |
| | Top-20 | $0.592 \pm 0.012$ | $0.828 \pm 0.052$ | $\mathbf{0.847 \pm 0.011}$ |
| | 50-th | $0.375 \pm 0.046$ | $0.465 \pm 0.039$ | $\mathbf{0.524 \pm 0.038}$ |
| D'Kitty Morphology | 100-th | $0.934 \pm 0.008$ | $\mathbf{0.950 \pm 0.041}$ | $0.912 \pm 0.017$ |
| | Top-5 | $0.912 \pm 0.005$ | $\mathbf{0.935 \pm 0.020}$ | $0.900 \pm 0.006$ |
| | Top-10 | $0.908 \pm 0.005$ | $\mathbf{0.918 \pm 0.009}$ | $0.894 \pm 0.006$ |
| | Top-20 | $\mathbf{0.901 \pm 0.010}$ | $0.892 \pm 0.028$ | $0.884 \pm 0.005$ |
| | 50-th | $\mathbf{0.887 \pm 0.007}$ | $0.792 \pm 0.016$ | $0.775 \pm 0.010$ |
| TF Bind 8 | 100-th | $0.843 \pm 0.046$ | $0.810 \pm 0.044$ | $\mathbf{0.946 \pm 0.043}$ |
| | Top-5 | $0.824 \pm 0.053$ | $0.790 \pm 0.021$ | $\mathbf{0.879 \pm 0.011}$ |
| | Top-10 | $0.742 \pm 0.042$ | $0.733 \pm 0.058$ | $\mathbf{0.832 \pm 0.018}$ |
| | Top-20 | $0.701 \pm 0.022$ | $0.696 \pm 0.062$ | $\mathbf{0.766 \pm 0.016}$ |
| | 50-th | $0.399 \pm 0.048$ | $0.355 \pm 0.011$ | $\mathbf{0.473 \pm 0.020}$ |
| TF Bind 10 | 100-th | $0.709 \pm 0.025$ | $0.703 \pm 0.022$ | $\mathbf{0.741 \pm 0.027}$ |
| | Top-5 | $0.626 \pm 0.040$ | $0.652 \pm 0.012$ | $\mathbf{0.681 \pm 0.021}$ |
| | Top-10 | $0.608 \pm 0.016$ | $0.613 \pm 0.011$ | $\mathbf{0.652 \pm 0.017}$ |
| | Top-20 | $0.588 \pm 0.013$ | $0.596 \pm 0.008$ | $\mathbf{0.621 \pm 0.009}$ |
| | 50-th | $0.487 \pm 0.039$ | $0.356 \pm 0.027$ | $\mathbf{0.491 \pm 0.006}$ |

*Table 11.* **Additional evaluation on RNA design tasks.** We extend our evaluation to three RNA optimization tasks (RNA-A/B/C) following the BootGen (Kim et al., 2023). These tasks involve designing length-14 RNA sequences to maximize transcription factor binding activity. Results report the normalized oracle score under different Top-$K$ metrics, where Top-$K$ denotes the average score of the top K generated candidates. We compare against two representative strong baselines, COMs and ExPT, from our original Design-Bench evaluation. The highest and second-highest scores per task are highlighted in **green** and blue , respectively.

| Task | Metric | COMs | ExPT | **DiBO**$_{(ours)}$ |
|---|---|---|---|---|
| RNA-A | 100-th | $0.362 \pm 0.011$ | $0.365 \pm 0.031$ | $\mathbf{0.395 \pm 0.015}$ |
| | Top-5 | $0.322 \pm 0.024$ | $0.331 \pm 0.014$ | $\mathbf{0.335 \pm 0.014}$ |
| | Top-10 | $0.305 \pm 0.020$ | $\mathbf{0.308 \pm 0.016}$ | $0.258 \pm 0.012$ |
| | Top-20 | $0.264 \pm 0.017$ | $\mathbf{0.294 \pm 0.025}$ | $0.224 \pm 0.007$ |
| | 50-th | $\mathbf{0.155 \pm 0.023}$ | $0.133 \pm 0.019$ | $0.141 \pm 0.004$ |
| RNA-B | 100-th | $\mathbf{0.350 \pm 0.063}$ | $0.348 \pm 0.011$ | $0.299 \pm 0.017$ |
| | Top-5 | $\mathbf{0.332 \pm 0.023}$ | $0.313 \pm 0.007$ | $0.229 \pm 0.009$ |
| | Top-10 | $\mathbf{0.299 \pm 0.030}$ | $0.292 \pm 0.006$ | $0.203 \pm 0.009$ |
| | Top-20 | $\mathbf{0.290 \pm 0.014}$ | $0.275 \pm 0.003$ | $0.177 \pm 0.059$ |
| | 50-th | $\mathbf{0.162 \pm 0.028}$ | $0.138 \pm 0.009$ | $0.087 \pm 0.004$ |
| RNA-C | 100-th | $0.445 \pm 0.018$ | $0.502 \pm 0.016$ | $\mathbf{0.528 \pm 0.037}$ |
| | Top-5 | $0.403 \pm 0.009$ | $\mathbf{0.487 \pm 0.013}$ | $0.413 \pm 0.015$ |
| | Top-10 | $0.392 \pm 0.031$ | $\mathbf{0.430 \pm 0.004}$ | $0.322 \pm 0.013$ |
| | Top-20 | $0.359 \pm 0.006$ | $\mathbf{0.415 \pm 0.007}$ | $0.280 \pm 0.009$ |
| | 50-th | $0.205 \pm 0.044$ | $\mathbf{0.247 \pm 0.023}$ | $0.162 \pm 0.007$ |

## A.7. Pretraining-Overlap Audit via Ranking Probing

Since DiBO is initialized from a pretrained diffusion LLM, a natural concern is whether the pretrained checkpoint already contains task-specific design-label knowledge from public benchmark assets. Because the pretraining corpus of LLADA-8B-INSTRUCT is not publicly available, an exact dataset-level overlap audit is not possible. We therefore perform a behavioral audit that probes whether the pretrained model can rank candidate designs according to their ground-truth labels before any domain adaptation or post-training.

For each task and each ranking size $K \in \{2, 3, 4, 5\}$, we sample 500 groups of $K$ candidate designs from the offline pool. The pretrained model is given only the rendered designs, without their labels, and is prompted to rank them from highest score to lowest score. The following probing prompt shows an example when $K = 3$, where each `<design>` is rendered in the same format as the corresponding benchmark task.

You are given 3 candidate designs.
Rank them from highest score to lowest score.
Return only a ranking over the option letters and no other text.

A: `<design A>`
B: `<design B>`
C: `<design C>`

Answer:

For each group, we enumerate all $K!$ possible answer rankings, such as `A > B > C`, `A > C > B`, and `B > A > C` when $K = 3$. We then compute the log probability assigned by the pretrained model to each possible ranking and sort all rankings by their model scores. Finally, we record the rank of the ground-truth ordering determined by the oracle labels. If the pretrained model already encodes benchmark-specific design-label knowledge, the correct ordering should be ranked close to 1; under random ranking, its expected rank is $(K! + 1)/2$ (e.g., 3.5 when $K = 3$).

*Table 12.* **Pretraining-overlap audit via ranking probing.** We test whether the pretrained LLADA-8B-INSTRUCT checkpoint encodes task-specific design-label ranking knowledge before any domain adaptation or post-training. For each task and each $K \in \{2, 3, 4, 5\}$, we sample 500 groups of $K$ designs, enumerate all $K!$ possible rankings, and score each ranking by the pretrained model. We report the average rank assigned to the ground-truth ordering, where lower is better. The random baseline is $(K! + 1)/2$.

| Task | $K = 2$ | $K = 3$ | $K = 4$ | $K = 5$ |
|---|---|---|---|---|
| Random Guess | 1.50 | 3.50 | 12.5 | 60.5 |
| Ant Morphology | 1.55 | 3.39 | 11.77 | 64.40 |
| D'Kitty Morphology | 1.55 | 3.30 | 11.84 | 60.90 |
| TF Bind 8 | 1.56 | 3.43 | 11.62 | 61.96 |
| TF Bind 10 | 1.58 | 3.29 | 11.62 | 63.00 |
| RNA-A | 1.59 | 3.49 | 11.85 | 62.94 |
| RNA-B | 1.56 | 3.50 | 11.51 | 63.54 |
| RNA-C | 1.54 | 3.52 | 11.58 | 64.44 |

As shown in Table 12, the average rank of the correct ordering is close to the random baseline across tasks and values of $K$. This suggests that the pretrained checkpoint does not exhibit meaningful prior knowledge of the design-label relationship.

