# OpenReview forum: "Training Diffusion Language Models for Black-Box Optimization"
_ICML.cc/2026/Conference — ICML 2026 spotlight_

### Official Review · Reviewer_ZFxf · 2026-03-08

**Soundness:** 3
**Presentation:** 3
**Significance:** 3
**Originality:** 2
**Overall Recommendation:** 5
**Confidence:** 4

**Summary:**

This paper presents an LLM-based approach to blackbox optimization. The authors propose a three-stage training methodology: starting with a pre-trained LLM, the "domain" of the LLM is adapted from natural language to a blackbox optimization setting (this is accomplished using natural language prompts with design and label delimiters); then using supervised fine tuning and reinforcement learning to hone the model's performance on the blackbox optimization task. One of the key details of the methodology is the use of diffusion LLMs instead, which (the authors claim) can capture bi-directional dependencies in the design problem better than autoregressive LLMs, and optimize over discrete token spaces and design spaces better than denoising diffusion-based blackbox optimization algorithms. The authors describe their approach in detail and demonstrate its performance in several experiments, which included comparisons to strong benchmarks and comprehensive ablation studies.

**Compliance With Llm Reviewing Policy:**

Affirmed.

**Key Questions For Authors:**

1. How does the choice to evenly sub-sample the experimental data sets sorted by label affect the experimental results?

2. What role does the pre-trained model play in the performance of the method?

3. What role does the uniform "offline policy" play in the reinforcement learning stage? Could other policies be better?

4. Which experiments demonstrate the importance of capturing bi-directional dependencies in design problems with diffusion LLMs?

I believe that discussing these points would strengthen the paper, especially question 4. Is this question addressed in other papers already?

**Limitations:**

The authors did not include an ablation study of the pre-trained LLM, so it is not clear what size role it plays in the success of the approach. It would be interesting to discuss this, as well as the general role of pre-training the LLM in the success of the framework. Do the authors suspect that pre-training gives the model better "priors" for solving design problems? (E.g., familiarity with robot locomotion or DNA binding affinity.) Or, does pre-training simply allow the model to ingest natural language prompts for blackbox optimization problems?

Given the large overlap between the approach of Yuan et al. (2026) and this work, I recommend that the authors move some of the discussion in Appendix A.5 into the main body of the paper.

I also recommend expanding the RL training section a bit. I think it is clear, but perhaps a bit too condensed. It would be more readable if usage of some of the RL jargon were explained in context.

**Strengths And Weaknesses:**

The proposed approach seems novel* and the experimental results demonstrate good performance of diffusion LLMs for blackbox optimization. The use of diffusion LLMs and the three-stage training methodology are well-motivated in the related work section and justified by the experimental results and ablation studies. The experiments include comparisons to strong benchmarks. The submission seems technically sound overall.

The paper is well-written and well-organized. Figure 1 provides a clear schematic of the authors' proposed approach. The experiments seem reasonable and the results are reported clearly and discussed thoroughly. I find the authors' assessment of their approach fair - in particular, they address the weak performance relative to other models on the D'Kitty task (their approach performed better on other tasks).

It seems that the authors have proposed a strong methodology for training diffusion LLMs to do blackbox optimization. The approach seems like it could be practically very useful for blackbox optimization tasks.

 *Yuan et al. (2026) seem to be the first to propose diffusion LLMs for blackbox optimization. Therefore, this paper cannot claim novelty for this detail of the approach. However, the authors emphasize (1) details of the training process and (2) training / adapting diffusion LLMs to do blackbox optimization, instead of using a frozen LLM as in Yuan et al. (2026). The authors present their claim to novelty fairly and carefully, and I do understand the important differences between their approach and that in Yuan et al. (2026). Therefore, I give a "fair" rating for originality.

---

> ### Author Rebuttal · Authors · 2026-03-31
>
> **Q1. Effect of evenly spaced sub-sampling.**
>
> We thank the reviewer for raising this important question. Our primary motivation for using evenly spaced sub-sampling was to ensure exact reproducibility and guarantee representative coverage across the entire label score spectrum.
>
> To verify this, we conducted an additional experiment comparing evenly spaced sub-sampling with random sub-sampling using the same pool size.
> As shown in [Table R4](https://anonymous.4open.science/r/Anonymous-dllm4bbo-D78A/rebuttal/Table_R4_R5.pdf), the performance differences between the two strategies are consistently small across all four tasks (within $\pm$ 0.01), and well within the standard deviation range.
>
> **Q2. Role of the pretrained model.**
>
> We evaluate the pretrained model directly before any domain adaptation or task-specific training. As shown in [Table R5](https://anonymous.4open.science/r/Anonymous-dllm4bbo-D78A/rebuttal/Table_R4_R5.pdf), the zero-shot pretrained model achieves weak performance on discrete tasks and fails to produce valid outputs on continuous tasks, indicating that pretraining alone is insufficient for solving the target optimization tasks.
>
> Furthermore, we verified that the pretrained LLaDA model does not implicitly memorize task-specific design-label rankings (i.e., no label leakage). We sampled 500 groups of $K=3$ candidate designs and prompted the model to rank them by their target property. We then computed the log probabilities for all $K!=6$ possible permutations. Across the 500 groups, the average rank of the ground-truth permutation was approximately 3.5, which perfectly aligns with random guessing.
>
> Conceptually, pretraining provides essential language capabilities and broad world priors (e.g., basic structural knowledge of DNA or robot kinematics) that allow the model to ingest natural language prompts. However, the precise design-to-label mappings must be explicitly injected during our domain adaptation and post-training stages.
>
> **Q3. Role of the uniform offline policy in RL.**
>
> We thank the reviewer for this insightful question.
>
> In standard PPO/GRPO training, the importance sampling denominator requires a known behavior policy (typically the language model that generated the responses) to provide token-level likelihoods.
>
> In contrast, the designs in our offline BBO dataset (e.g., DNA sequences or robot morphologies) originate from external, real-world processes rather than a known model. Consequently, the true data-generating distribution is inaccessible, and we lack an explicit behavior policy to provide likelihood estimates.
>
> To address this, we adopt a uniform offline policy as a simple and stable reference assumption, which closely follows recent off-policy LLM-RL practices (Fu et al., 2025; Yan et al., 2025).
>
> **Q4. Which experiments demonstrate the importance of capturing bi-directional dependencies?**
>
> While the original manuscript included OPRO (Yang et al., 2023) as an out-of-the-box autoregressive (AR) baseline, we agree that a strictly controlled ablation is necessary to isolate the architectural contribution.
>
> Per suggestion, we conducted a controlled architecture swap. We applied our exact training framework (identical prompt construction, offline dataset, delimiter tokens, context strategy, and DA → SFT → RL pipeline) to an AR backbone (LLaMA-3.1-8B-Instruct), replacing only the diffusion model.
>
> As shown in the newly added [Table R1](https://anonymous.4open.science/r/Anonymous-dllm4bbo-D78A/rebuttal/Table_R1.pdf), the diffusion backbone consistently outperforms the matched AR backbone across all four tasks and at every training stage. This systematic improvement isolates the role of the architecture, providing direct empirical evidence that the bidirectional refinement of diffusion models is fundamentally better suited for capturing the complex interdependencies in structured design problems than left-to-right AR generation.
>
> **Q5. Comparison with Yuan et al. (2026).**
>
> We thank the reviewer for this suggestion. To clarify the differences in problem settings and optimization regimes, we will add the following paragraph to the end of Section 4.5 (Line 420):
>
> **Comparison with inference-only diffusion optimization.**
> Yuan et al. (2026) explore inference-only diffusion optimization using a frozen model combined with Monte Carlo Tree Search (MCTS). In contrast, our approach actively adapts the diffusion LLM via domain adaptation, supervised fine-tuning, and reinforcement learning. Furthermore, their method targets an extremely few-shot regime (e.g., $n_{\text{pool}} = 10$), whereas our setting leverages substantially larger offline datasets (e.g., $n_{\text{pool}} = 500$). Given these distinct operating conditions, we exclude their method from our main baselines, but we view the two approaches as complementary.
>
> **Q6. Expanding RL training section.**
>
> Per your suggestion, we would like to add our response in Q3 to the paragraph around line 200.

---

> > ### Author Rebuttal · Reviewer_ZFxf · 2026-04-02
> >
> > The authors have addressed my concerns. I retain my favorable assessment of the paper.

---

> > > ### Author Response · Authors · 2026-04-04
> > >
> > > Thank you for your continued support and favorable assessment of our paper. We are glad to hear that our rebuttal successfully addressed your concerns. We appreciate the time and effort you dedicated to reviewing our work.

---

### Official Review · Reviewer_ur6w · 2026-03-09

**Soundness:** 3
**Presentation:** 3
**Significance:** 3
**Originality:** 3
**Overall Recommendation:** 5
**Confidence:** 4

**Summary:**

This paper introduces a diffusion model that is trained to do black-box optimization, including over input spaces that have traditionally been challenging to model. It shows that diffusion models are a more powerful model for this setting as they do not have the autoregressive assumption that models like GPT have, and so can optimize in more intricately constrained design spaces. The paper also introduces a training process that incorporates domain adaptation (to learning the right representation), supervised fine tuning, and RL (to focus the model on the task of optimal sequence reconstruction). These allow the model to outperform a broad set of alternative approaches when optimizing in complex design spaces.

**Compliance With Llm Reviewing Policy:**

Affirmed.

**Final Justification:**

The rebuttal addressed my main concern. The paper is novel and using interesting methodology to solve an important problem, and to me is a good fit for ICML.

**Key Questions For Authors:**

1. What is the GP kernel?
2. Please describe better the prompt template variation study and explain why the extraneous phrase "You are a helpful optimization assistant tasked with..." if the results are not sensitive to the particular language in the prompt?

**Limitations:**

yes

**Strengths And Weaknesses:**

Strengths:
1. The use of diffusion models for BBO is novel, and is well-motivated.
2. The paper clearly describes the training process, experiments, and results.
3. The set of baselines used for the experiments is rather broad, and the experiments are sufficiently diverse.
4. The paper includes a thorough set of ablations that provide individual tests of the main contributions of the paper.
5. The model performs well compared to baselines, and the paper is able to identify some of the reasons it performed well in the context of the motivation given for the method.

Overall the paper provides a novel and seemingly useful approach to a challenging problem, and the ablations provide insight and understanding that I expect can be used more generally.

Weaknesses:
1. The paper does not specify what GP kernel was used as a baseline. These are high-dimensional problems so it should be a kernel suitable for high-dimensional problems, such as that of Hvarfner et al. ICML 2024.
2. The paper shows that the presence of the delimiter token is very important for model performance, but does not explore more what attributes of the delimiter are important. Is it just important that there be any delimiter (e.g. "&" or "|") or is there value specifically to the semantic tokens "design" and "label"?
3. The paper does a study of prompt template variation to claim that the results are not sensitive to the particular prompt, but no details are given on this. How was this done, by asking GPT to write the existing prompt? There must not be too much variation in the prompts, as it seems like it would be an easy feat to break the model by choosing a really bad prompt. If the particular prompt text doesn't matter, then why does the prompt include the extraneous phrase "You are a helpful optimization assistant tasked with..."?

---

> ### Author Rebuttal · Authors · 2026-03-31
>
> **Q1. GP kernel.**
>
> We thank the reviewer for pointing out this missing detail.
> For the GP kernel, we use a standard Gaussian Process with a squared-exponential (Radial Basis Function, RBF) kernel.
> This is a widely adopted default choice in black-box optimization and surrogate modeling, and is consistent with prior BBO works such as ExPT (Nguyen et al., 2023), which also employs an RBF kernel for its GP-based components.
>
> We agree that kernel design can play an important role in high-dimensional settings.
> Our intention here was to include a representative and commonly used GP kernel under the same benchmark protocol as prior work.
> We will explicitly state the kernel choice in the revision to improve clarity and reproducibility.
>
> **Q2. Delimiter token.**
>
> We thank the reviewer for this insightful question.
> In our implementation, delimiter tokens such as `|design-start|` and `|label-start|` are introduced as atomic special tokens in the tokenizer, rather than being decomposed into separate lexical units (e.g., `design` or `start`).
> As a result, their primary role is to provide explicit structural boundaries between fields, rather than to convey semantic meaning through the words themselves. Because they are treated as newly initialized unique IDs in the embedding layer, using alternative strings for these atomic tokens (e.g., defining them as `|sample-start|` or `|score-start|` during vocabulary expansion) would yield mathematically and functionally identical model behavior.
>
> **Q3. Prompt template variation.**
>
> We thank the reviewer for requesting clarification on this point.
> Our prompt-variation study was conducted by generating multiple paraphrased prompt templates using ChatGPT-5.2.
> These templates were designed to preserve the same task semantics and structured format (design/label separation), while varying the surface-level natural-language phrasing.
>
> Specifically, the created prompts consist of reasonable paraphrastic instruction variants rather than adversarial or intentionally degraded prompts.
> Only one template (the example shown in the appendix) begins with ``You are a helpful optimization assistant'', while other templates include completely different instruction styles. We take TF Bind 8 for instance:
>
> - "The goal is to design an 8-length DNA sequence with maximum binding affinity to the transcription factor ..."
> - "Below is a reference dataset of DNA sequences ranked by binding affinity to the transcription factor ..."
> - "Optimization Task: Find a DNA sequence (length = 8) that binds more strongly to ..."
>
> Our goal in this study was to test robustness to natural paraphrasing commonly encountered in instruction-tuned settings, rather than robustness to arbitrary or adversarial prompt choices.
> We therefore do not claim invariance to poorly specified prompts.
> All templates preserve the same structured delimiter tokens and output format, ensuring that the variation is limited to surface-level wording rather than structural specification.
>
> Regarding the phrase "You are a helpful optimization assistant...'':
> this prefix appears in only one template and was included as a natural instruction-style phrasing compatible with instruction-tuned backbones, rather than as a uniquely necessary component.
> Empirically, templates without this phrase achieved comparable performance, indicating that the results are not dependent on this specific wording.
> We will clarify these details and provide additional prompt examples in the revision.

---

> > ### Author Rebuttal · Reviewer_ur6w · 2026-04-03
> >
> > I thank the authors for their response.
> >
> > Q1: The rebuttal explains that the GP baseline uses a GP with an RBF kernel as the baseline. That is an inappropriate baseline for problems of this dimensionality. The rebuttal claims that "Our intention here was to include a representative and commonly used GP kernel "; that is neither representative nor commonly used in the high-dimensional setting that is the focus of this paper. Implementations of high-dimensional kernels (e.g. that of Hvarfner et al. ICML 2024) are widely available and could be used with little extra difficulty, as they do not require any change other than to swap in the new kernel. See for instance https://github.com/meta-pytorch/botorch/discussions/2451 .

---

> > > ### Author Response · Authors · 2026-04-04
> > >
> > > **Follow up on Q1.**
> > >
> > > We thank the reviewer for pointing out this important issue.
> > > After carefully reviewing the suggested paper and the implementation guidance, we agree that kernel strategies tailored for high-dimensional settings are very important.
> > >
> > > Per your suggestion, we replaced the original RBF kernel with the high-dimensional GP formulation using dimension-scaled priors as described in Hvarfner et al. (ICML 2024), and update the results on baselines using GP kernels (i.e. Grad-mean and Grad-EI).
> > >
> > > The updated results are reported below. We report both the maximum and median scores, and use $\Delta$ to denote the performance improvement.
> > >
> > > *(a) Maximum Scores*
> > > | Method | Kernel | Ant Morphology | D'Kitty Morphology | TF Bind 8 | TF Bind 10 |
> > > |-------|--------|----------------|--------------------|------------|------------|
> > > | **Grad-mean** | RBF | 0.644 ± 0.039 | 0.907 ± 0.016 | 0.666 ± 0.011 | 0.695 ± 0.027 |
> > > |  | High-dim | **0.709 ± 0.002** | 0.920 ± 0.008 | 0.843 ± 0.082 | **0.736 ± 0.016** |
> > > |  | Δ | +0.065 ↑ | +0.013 ↑ | +0.177 ↑ | +0.041 ↑ |
> > > | **Grad-EI** | RBF | 0.626 ± 0.002 | 0.901 ± 0.045 | 0.673 ± 0.012 | 0.689 ± 0.013 |
> > > |  | High-dim | 0.655 ± 0.002 | **0.923 ± 0.010** | **0.864 ± 0.091** | 0.727 ± 0.024 |
> > > |  | Δ | +0.029 ↑ | +0.022 ↑ | +0.191 ↑ | +0.038 ↑ |
> > > | **DiBO (ours)** | -- | **0.944 ± 0.016** | **0.923 ± 0.002** | **0.965 ± 0.038** | **0.755 ± 0.012** |
> > > |||||||
> > >
> > > *(b) Median Scores*
> > > | Method | Kernel | Ant Morphology | D'Kitty Morphology | TF Bind 8 | TF Bind 10 |
> > > |-------|--------|----------------|--------------------|------------|------------|
> > > | **Grad-mean** | RBF | 0.351 ± 0.002 | 0.879 ± 0.015 | 0.623 ± 0.030 | 0.394 ± 0.004 |
> > > |  | High-dim | 0.472 ± 0.013 | **0.858 ± 0.003** | **0.441 ± 0.008** | **0.461 ± 0.008** |
> > > |  | Δ | +0.121 ↑ | −0.021 ↓ | −0.182 ↓ | +0.067 ↑ |
> > > | **Grad-EI** | RBF | 0.368 ± 0.046 | 0.872 ± 0.004 | 0.595 ± 0.019 | 0.453 ± 0.047 |
> > > |  | High-dim | **0.504 ± 0.013** | 0.849 ± 0.008 | 0.436 ± 0.004 | 0.459 ± 0.024 |
> > > |  | Δ | +0.136 ↑ | −0.023 ↓ | −0.159 ↓ | +0.006 ↑ |
> > > | **DiBO (ours)** | -- | **0.543 ± 0.028** | **0.837 ± 0.003** | **0.599 ± 0.061** | **0.504 ± 0.012** |
> > > |||||||
> > >
> > > We observe that adopting the high-dimensional kernel improves the GP-based baselines in most of the cases, confirming that kernel choice is important in high-dimensional settings.
> > >
> > > Importantly, even with this stronger kernel configuration, our method remains competitive and achieves superior performance on the majority of cases, despite D'Kitty.
> > >
> > > We sincerely thank the reviewer for highlighting this issue.
> > > In the final revision, we will explicitly document the kernel configuration and update the two GP-based baseline results accordingly to ensure clarity and reproducibility.

---

### Official Review · Reviewer_w2ve · 2026-03-12

**Soundness:** 3
**Presentation:** 2
**Significance:** 3
**Originality:** 2
**Overall Recommendation:** 4
**Confidence:** 2

**Summary:**

This paper proposes DiBO, a framework that adapts diffusion language models for offline black-box optimization (BBO). The core idea is that autoregressive LLMs are limited by their unidirectional generation, whereas diffusion LLMs can capture bidirectional dependencies that are otherwise ignored in domains like DNA sequences. The model begins by unifying heterogeneous BBO signals through explicit delimiter tokens. Domain adaptation is achieved via joint masked-token prediction over prompts and responses. The model is then further aligned with high-quality label designs through supervised fine-tuning (SFT) and reinforcement learning (RL) guided by label improvements.

**Compliance With Llm Reviewing Policy:**

Affirmed.

**Final Justification:**

Authors have addressed most of my concerns.

**Key Questions For Authors:**

1. The bidirectionality claim does not appear to be experimentally validated. Is there a direct ablation comparing the diffusion LLM against a traditional autoregressive LLM under the same training pipeline? Without this, how do the authors justify bidirectionality as a core contribution?
2. The log-probability approximation is insufficiently justified. Could the authors provide more rigorous theoretical or empirical justification for these choices?
3. Failure cases across tasks deserve deeper analysis. What explains the model's exceptional performance on some Design-Bench tasks while struggling on others such as D'Kitty? Understanding these discrepancies is important for characterizing the model's strengths and limitations.
4. Why was only the 100th percentile (50th percentile listed in appendix) normalized oracle score used for evaluation? Given the limited set of Design-Bench benchmarks, would the authors consider reporting complementary metrics such as top-K average, or success rate for a more complete and honest assessment?

**Limitations:**

Yes

**Strengths And Weaknesses:**

The authors' argument that bidirectional modeling is better suited than autoregressive generation for design problems with non-sequential dependencies is intuitive. However, the research has few notable gaps like the narrow benchmark scope and the lack of a controlled architecture comparison to truly validate the bidirectionality claim.

---

> ### Author Rebuttal · Authors · 2026-03-31
>
> **Q1: Bidirectionality and matched autoregressive comparison.**
>
> While the original manuscript included OPRO (Yang et al., 2023) as an out-of-the-box autoregressive (AR) baseline, we agree that a strictly controlled ablation is necessary to isolate the architectural contribution.
>
> Per your suggestion, we conducted a controlled architecture swap. We applied our exact training framework (identical prompt construction, offline dataset, delimiter tokens, context strategy, and DA → SFT → RL pipeline) to an AR backbone (LLaMA-3.1-8B-Instruct), replacing only the diffusion model.
>
> As shown in the newly added [Table R1](https://anonymous.4open.science/r/Anonymous-dllm4bbo-D78A/rebuttal/Table_R1.pdf), the diffusion backbone consistently outperforms the matched AR backbone across all four tasks and at every training stage. This systematic improvement isolates the role of the architecture, providing direct empirical evidence that the bidirectional refinement of diffusion models is fundamentally better suited for capturing the complex interdependencies in structured design problems than left-to-right AR generation.
>
> **Q2: Justification of the log-probability approximation.**
>
> We thank the reviewer for raising this point. The mean-field approximation of sequence log probability in Eq.~(6) follows prior diffusion-LM post-training work (Zhao et al., 2025), and is adopted to provide a tractable approximation for sequence likelihood during RL optimization.
>
> Zhao et al. (2025) provide empirical evidence showing that this approximation yields stable optimization and strong performance in diffusion-LM post-training, supporting its validity as a practical approximation.
>
> **Q3: Failure cases and task discrepancy analysis.**
>
> We thank the reviewer for this valuable question. We analyzed the underlying offline datasets and found that D'Kitty exhibits a highly bimodal label distribution. The samples are heavily concentrated at the extreme high and low ends, with very few median-scoring designs. In contrast, the other three tasks (Ant, TF8, TF10) feature smooth, continuous label distributions.
>
> This bimodal nature severely limits the transition states available for the model to learn a continuous optimization trajectory. To further validate our model's robustness and generalizability across diverse problem landscapes, we have extended our evaluation to three new RNA optimization tasks. Please see Q4 for these detailed results and analysis.
>
> **Q4: Evaluation metrics and complementary summaries.**
>
> We initially reported the 100th and 50th percentile normalized oracle scores to strictly align with established offline BBO evaluation protocols, such as those used in COMs (Trabucco et al., 2021) and RoMA (Yu et al., 2021). We agree that complementary metrics provide a more comprehensive assessment. We have expanded our evaluation to include Top-$K$ average scores (specifically Top-5, Top-10, and Top-20). As shown in the newly added [Table R3 (b)](https://anonymous.4open.science/r/Anonymous-dllm4bbo-D78A/rebuttal/Table_R3.pdf), DiBO consistently outperforms strong baselines across these Top-$K$ metrics on most of Design-Bench tasks despite D'Kitty. This confirms that our method's performance gains are not isolated to extreme best-case outcomes, but reflect a systematically higher-quality candidate pool overall.
>
> Finally, to further broaden the evaluation scope, we also introduce three additional RNA optimization tasks (RNA-A/B/C) following BootGen (Kim et al., 2023). These tasks involve designing length-14 RNA sequences to maximize transcription factor binding activity, representing a new biological sequence design domain beyond the original Design-Bench settings. As shown in [Table R3 (a)](https://anonymous.4open.science/r/Anonymous-dllm4bbo-D78A/rebuttal/Table_R3.pdf), DiBO maintains strong performance across most of these new tasks, further supporting the generalizability of the proposed approach.

---

> > ### Author Rebuttal · Reviewer_w2ve · 2026-04-04
> >
> > The authors have addressed my concerns. I will update my response accordingly.

---

> > > ### Author Response · Authors · 2026-04-04
> > >
> > > We are glad to hear that our clarifications and the new experiments have fully resolved your concerns. We appreciate your constructive feedback throughout this process and look forward to your updated assessment.

---

### Official Review · Reviewer_mz9W · 2026-03-13

**Soundness:** 2
**Presentation:** 2
**Significance:** 2
**Originality:** 2
**Overall Recommendation:** 4
**Confidence:** 2

**Summary:**

This paper proposes DiBO, a method for adapting a pretrained diffusion LLM to offline black-box optimization by converting BBO into a unified prompt–response format with delimiter tokens, followed by domain adaptation, supervised fine-tuning, and reinforcement learning. The paper argues that diffusion LLMs are attractive because design problems often exhibit bidirectional dependencies, and it evaluates the method on four Design-Bench tasks in a small-data regime with n_{\text{pool}}=500. Empirically, DiBO is strong: it achieves the best mean/median rank overall and outperforms the strongest baseline on 3 of 4 reported task

**Compliance With Llm Reviewing Policy:**

Affirmed.

**Final Justification:**

The additional experiments provided in the rebuttal addressed my main concerts and as result I have increased it to weak accept.

**Key Questions For Authors:**

1) Can the authors provide a matched non-diffusion baseline using the same prompt construction and the same DA/SFT/RL pipeline, to isolate whether diffusion is actually the decisive
2) Can the authors discuss or audit possible pretraining overlap between LLADA-8B-Instruct and public Design-Bench assets? Ideally would there should be evaluations done with datasets released after the weights of LLADA
3) Why is the evaluation limited to only four Design-Bench tasks? Could the authors extend to additional tasks to better support the paper’s generality claims?

**Limitations:**

yes

**Strengths And Weaknesses:**

Strengths:
The main strength is empirical performance. On the reported Design-Bench setup, DiBO is the strongest method overall and wins on Ant, TFBind8, and TFBind10, while remaining competitive on D’Kitty. This also shows that the method is competitive across different modalities. Additionally


Limitations:
1) The evaluation scope is narrow. The paper evaluates only on four Design-Bench tasks, all in the same small-data setup with a fixed offline pool of 500 samples
2) The paper does not explore empirically how much diffusion is responsible for the performance. The motivation of the paper is that diffusion helps because of bidirectional modeling, and the introduction repeatedly contrasts this with autoregressive generation. However, the ablations never perform a controlled architecture swap where the same prompt construction, same data pipeline, and same post-training procedure are used with a matched non-diffusion backbone.
3)  The model is initialized from the pretrained diffusion LLM LLADA-8B-Instruct. It does not address whether public benchmark assets or underlying task content may already have been present in the foundation model’s pretraining data.
4) The inference procedure is not clearly detailed in the paper. The paper explains generic diffusion-LM generation as iterative denoising from a fully masked sequence, and the RL section uses a one-step unmasking approximation for training.  But there is still no precise task-level test-time algorithm: the paper does not clearly say how contexts are chosen at inference, whether they are fixed or resampled across the 128 candidates, what exact stochastic decoding choices are used, or how output parsing/validation is handled for structured designs.

---

> ### Author Rebuttal · Authors · 2026-03-31
>
> **Q1: Matched non-diffusion baseline.**
>
> While the original manuscript included OPRO (Yang et al., 2023) as an out-of-the-box autoregressive (AR) baseline, we agree that a strictly controlled ablation is necessary to isolate the architectural contribution.
>
> Per your suggestion, we conducted a controlled architecture swap. We applied our exact training framework (identical prompt construction, offline dataset, delimiter tokens, context strategy, and DA → SFT → RL pipeline) to an AR backbone (LLaMA-3.1-8B-Instruct), replacing only the diffusion model.
>
> As shown in the newly added [Table R1](https://anonymous.4open.science/r/Anonymous-dllm4bbo-D78A/rebuttal/Table_R1.pdf), the diffusion backbone consistently outperforms the matched AR backbone across all four tasks and at every training stage. This systematic improvement isolates the role of the architecture, providing direct empirical evidence that the bidirectional refinement of diffusion models is fundamentally better suited for capturing the complex interdependencies in structured design problems than left-to-right AR generation.
>
> **Q2: Possible pretraining overlap.**
>
> We agree that pretraining contamination is a valid concern. Because LLaDA developers [have not publicly released their pretraining data or framework](https://github.com/ML-GSAI/LLaDA?tab=readme-ov-file#pre-training-and-supervised-fine-tuning), an exact dataset-level audit is not possible. To address this, we conducted a behavioral audit to determine whether the pretrained LLaDA model implicitly encodes task-specific design-label ranking knowledge.
>
> We sample 500 groups of $K=3$ candidate designs. For example, we provide three DNA sequences (A, B, and C) and prompt the model: *"You are given 3 candidate DNA sequence designs. Rank them from highest affinity to lowest affinity to the protein. Return only a ranking over the option letters and no other text."* We then compute the log probabilities for all $K!=6$ possible permutations (e.g., $B>A>C$, $C>B>A$, etc.). If the true ground-truth ranking is $B>C>A$, we check where this correct permutation ranks among the six options based on the model's assigned probabilities.
>
> Across the 500 groups, the average rank of the correct permutation is approximately 3.5, which aligns well with random guessing. We expanded this audit to include $K=2,4,5$ (detailed in [Table R2](https://anonymous.4open.science/r/Anonymous-dllm4bbo-D78A/rebuttal/Table_R2.pdf)), and the results consistently reflect random chance.
>
> This indicates that the pretrained model does not exhibit meaningful knowledge of the design-label relationships. While we cannot fully rule out data overlap without original corpus access, this audit strongly suggests that our improvements stem from the proposed training pipeline rather than memorized benchmark knowledge.
>
> **Q3: Evaluation scope and generality.**
>
> To address concerns about evaluation scope and demonstrate DiBO's generality, we have expanded our experiments to include three additional RNA optimization tasks (RNA-A, B, C) adapted from BootGen (Kim et al., 2023). These tasks, which involve designing length-14 RNA sequences to maximize transcription factor binding, introduce a new biological sequence domain distinct from Design-Bench.
>
> As shown in [Table R3](https://anonymous.4open.science/r/Anonymous-dllm4bbo-D78A/rebuttal/Table_R3.pdf), DiBO outperforms strong baselines (ExPT and COMs) in most cases. This advantage holds across both maximum and median scores, as well as multiple Top-K metrics (the average score of the top K generated candidates).
>
> Regarding the offline pool size, the strict 500-sample limit is a deliberate design choice reflecting the primary challenge of real-world offline BBO. In domains like biology and chemistry, label acquisition through physical experimentation is highly cost-prohibitive. The core objective of this field is to discover improved candidates from a severely restricted set of evaluated samples. Evaluating our method under these tight data constraints aligns directly with established offline BBO methodologies, such as the few-shot settings formalized by ExPT (Nguyen et al., 2023).
>
> **Q4: Inference procedure clarification.**
>
> We thank the reviewer for pointing this out. The full inference pipeline shared by all tasks is clarified below.
>
> For each candidate, we randomly sample a given number of context examples (7 by default) from the offline pool. The context is not fixed but randomly resampled, and is sampled independently for each candidate.
>
> We then perform masked response completion using a single forward pass with greedy token filling. No stochastic sampling (e.g., temperature) is used during token generation. Duplicate outputs are discarded, and the generation process continues until obtaining 128 valid unique candidates.
>
> After domain adaptation (DA), the model reliably produces outputs that follow the required structured format, so no additional format-specific validation is required.

---

> > ### Author Rebuttal · Reviewer_mz9W · 2026-04-01
> >
> > the rebuttal substantially improves confidence by adding a controlled AR baseline and broader evaluation. So I am happy to increase my score to weak accept.

---

> > > ### Author Response · Authors · 2026-04-01
> > >
> > > Thank you for your prompt reply and for confirming that our additional baselines and evaluations have fully resolved your concerns. We are glad to hear it.
> > >
> > > We deeply appreciate your constructive feedback throughout the review process and are grateful for your support in raising your evaluation to a Weak Accept.

---

### Decision · Program_Chairs · 2026-04-30

**Decision:**

Accept (spotlight)

**Comment:**

This submission introduces DiBO, a framework for adapting diffusion language models to offline black-box optimization via a multi-stage pipeline consisting of domain adaptation, supervised fine-tuning, and reinforcement learning. During the initial review phase, concerns were raised regarding the narrow scope of the evaluation, potential data leakage from large-scale pretraining, and the lack of a controlled comparison between diffusion and autoregressive architectures to justify the core motivation. The authors provided a comprehensive and professional rebuttal that effectively addressed these points by incorporating a matched-architecture ablation using a LLaMA-3.1 backbone, which isolated the benefits of bidirectional modeling, and by performing a behavioral audit that mitigated concerns regarding benchmark contamination. Furthermore, the expansion of the experimental suite to include additional RNA optimization tasks and the adoption of stronger high-dimensional GP kernels significantly strengthened the empirical evidence. The reviewers reached a unanimous consensus that the authors' responses were thorough and that the technical rigor of the manuscript was high. While the work is technically sound and the results are well-validated, the contribution is characterized as a solid and well-executed advancement within the specific sub-area of black-box optimization. Therefore, the Area Chair recommends acceptance as a solid contribution to the program, noting that the manuscript now meets the high standards of ICML following the productive discussion period.